# GPS++: Reviving the Art of Message Passing for Molecular Property Prediction

**Dominic Masters[1]\*,  Josef Dean[1]\*,  Kerstin Klaser[1]\*,  Zhiyi Li[1],  Sam Maddrell-Mander[1],**
**Adam Sanders[1],  Hatem Helal[1],  Deniz Beker[1],  Andrew Fitzgibbon[1],  Shenyang Huang[2, 3, 4],**
**Ladislav Rampášek[3, 5], Dominique Beaini[2, 3, 5]**

[1]Graphcore    [2]Valence    [3]Mila - Québec AI Institute    [4]McGill University    [5]Université de Montréal

**Reviewed on OpenReview:** `https://openreview.net/forum?id=moVEUgJaHO`

## Abstract

We present GPS++, a hybrid Message Passing Neural Network / Graph Transformer model for molecular property prediction. Our model integrates a well-tuned local message passing component and biased global attention with other key ideas from prior literature to achieve state-of-the-art results on large-scale molecular dataset PCQM4Mv2. Through a thorough ablation study we highlight the impact of individual components and find that nearly all of the model's performance can be maintained without any use of global self-attention, showing that message passing is still a competitive approach for 3D molecular property prediction despite the recent dominance of graph transformers. We also find that our approach is significantly more accurate than prior art when 3D positional information is not available.

Table 1: Comparison of model size and accuracy on large-scale molecular property prediction dataset PCQM4Mv2.

| Model | # Params | Model Type | Validation MAE (meV) ↓ |
|---|---|---|---|
| GIN-virtual (Hu et al., 2021) | 6.7M | MPNN | 108.3 |
| GPS (Rampášek et al., 2022) | 19.4M | Hybrid | 85.8 |
| GEM-2 (Liu et al., 2022a) | 32.1M | Transformer | 79.3 |
| Global-ViSNet (Wang et al., 2022b) | 78.5M | Transformer | 78.4 |
| Transformer-M  (Luo et al., 2022) | 69.0M | Transformer | 77.2 |
| GPS++ [MPNN only] | 40.0M | MPNN | 77.2 |
| **GPS++** | **44.3M** | **Hybrid** | **76.6** |

## 1   Introduction

Among many scientific areas, deep learning is having a transformative impact on molecular property prediction tasks for biological and chemical applications (Keith et al., 2021; Reiser et al., 2022). In particular, the aim is to replace or augment expensive and/or time-consuming experiments and first-principles methods with more efficient machine learning models. While there is a long history of machine learning methods in this field, two particular approaches have been dominant as of late for processing graph-structured molecular data: message passing neural networks (MPNNs) iteratively build graph representations of molecules by sending information explicitly along edges defined by bonds (Gilmer et al., 2017; Battaglia et al., 2018); while graph transformers treat the nodes (atoms) as all-to-all connected and employ global self-attention approaches (Vaswani et al., 2017), optionally building in local inductive biases through augmented attention (Ying et al., 2021a; Luo et al., 2022). While transformers have been extremely successful in other domains, the quadratic complexity

---

\*Equal Contribution

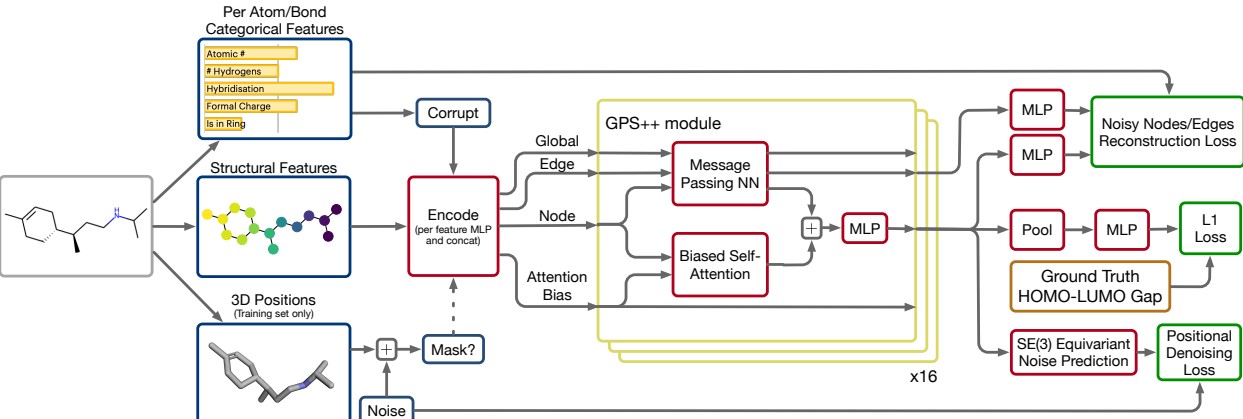

Figure 1: Augmented General Powerful Scalable (GPS++) Graph Transformer overview. (left) GPS++ takes on the input a featurised molecular graph. Chemical, positional, and structural node (atom) and edge (bond) features are detailed in section 4.2. Geometric 3D information is provided only optionally during training, e.g. inference on PCQM4Mv2 test set needs to be done without their explicit knowledge. (middle) A stack of GPS++ modules that combine a custom local MPNN and a global biased self-attention mechanism (akin to Transformer-M) to learn expressive representations, see section 4.1. (right) Global graph pooling with final prediction head, and auxiliary denoising tasks, see section 5.2.

with respect to the number of nodes is a significant obstacle for larger molecules. This makes MPNNs an attractive approach due to their linear scaling with graph size, however, issues like oversmoothing (Li et al., 2018), oversquashing, and underreaching (Alon & Yahav, 2021) have been found to limit their effectiveness.

In this work we focus on the task of predicting the HOMO-LUMO energy gap, an important quantum chemistry property being the minimum energy needed to excite an electron in the molecular structure. This property is typically calculated using Density Functional Theory (DFT) (Kohn & Sham, 1965), the *de facto* method used for accurately predicting quantum phenomena across a range of molecular systems. Unfortunately, traditional DFT can be extremely computationally expensive, prohibiting the efficient exploration of chemical space (Dobson, 2004), with some approaches taking more than 8 hours per molecule per CPU (Axelrod & Gomez-Bombarelli, 2022). Within this context the motivation for replacing it with fast and accurate machine learning models is clear. While this task does aim to accelerate the development of alternatives to DFT it also serves as a proxy for other molecular property prediction tasks. Therefore, it can potentially benefit a range of scientific applications in fields like computational chemistry, material sciences and drug discovery.

The PubChemQC project (Nakata & Shimazaki, 2017) is one of the largest widely available DFT databases, and from it is derived the PCQM4Mv2 dataset, released as a part of the Open Graph Benchmark Large Scale Challenge (OGB-LSC) (Hu et al., 2021), which has served as a popular testbed for development and benchmarking of novel graph neural networks (GNNs). The original OGB-LSC 2021 competition motivated a wide range of solutions using both MPNNs and transformers. However, following the success of the winning method Graphormer (Ying et al., 2021a), subsequent work has resulted in a large number of graph transformer methods for this task with comparatively little attention given to the message passing methods that have been successful in other graph-structured tasks.

In this work we build on the work of Rampášek et al. (2022) that advocates for a hybrid approach, including both message passing and transformer components in their General, Powerful, Scalable (GPS) framework. Specifically, we build GPS++, which combines a large and expressive message-passing module with a biased self-attention layer to maximise the benefit of local inductive biases while still allowing for effective global communication. Furthermore, by integrating a grouped input masking method (Luo et al., 2022) to exploit available 3D positional information and carefully crafting a range of diverse input features we achieve the best-reported result on the PCQM4Mv2 validation data split of 76.6 meV mean absolute error (MAE).

Next, we perform an extensive ablation study to understand the impact of different model components on the performance. Surprisingly, we find that even without a global self-attention mechanism (seen in graph

transformer architectures), extremely competitive performance can be achieved. Therefore, we argue that MPNNs remain viable in the molecular property prediction task and hope the results presented in this work can spark a renewed interest in this area. We also observe that when solely focusing on 2D molecular data (without 3D conformer coordinates), our proposed GPS++ model significantly outperforms other such works.

Our contributions can be summarised as follows:

- We show that our hybrid MPNN/Transformer model, GPS++, is a parameter-efficient and effective approach to molecular property prediction, achieving state-of-the-art MAE scores for PCQM4Mv2 even when compared to parametrically larger models.

- We find that even without self-attention, our model matches the performance of prior state-of-the-art transformers, highlighting that well-optimised MPNNs are still highly effective in this domain.

- We investigate how different model components affect the model performance, in particular highlighting the impact of the improved chemical feature choice, 3D positional features, structural encodings and architectural components.

**Reproducibility:** Source code to reproduce our results can be found at: `https://github.com/graphcore/ogb-lsc-pcqm4mv2`.

## 2 Related Work

Before the advent of deep learning for graph representation learning (Bronstein et al., 2021), molecular representation in chemoinformatics rested on feature engineering of descriptors or fingerprints (Rogers & Hahn, 2010; Keith et al., 2021). After learnable fingerprints (Duvenaud et al., 2015) the general framework of GNNs, often based on MPNNs, has been rapidly gaining adoption (Reiser et al., 2022). Not all approaches follow the MPNN model of restricting compute to the sparse connectivity graph of the molecule, for example the continuous-filter convolutions in SchNet (Schütt et al., 2018) and the transformer-based Grover (Rong et al., 2020) employ dense all-to-all compute patterns.

GNN method development has also been facilitated by the availability of well motivated (suites of) benchmarking datasets such as QM9 (Ramakrishnan et al., 2014), MoleculeNet (Wu et al., 2018), Open Graph Benchmark (OGB) (Hu et al., 2020), OGB-LSC PCQM4Mv2 (Hu et al., 2021), Therapeutics Data Commons (Huang et al., 2022) for molecular property prediction or MD17 (Chmiela et al., 2017), ANI-1 (Smith et al., 2017), Open Catalyst (Tran et al., 2022) for structural conformers or molecular force field predictions.

Depending on the nature of the task, the geometric 3D structure of input molecules needs to be taken into account. This imposes an important constraint on the models, that need to be roto-translation invariant in case of global molecular property prediction and roto-translation equivariant in case of conformer or force field prediction, in addition to input permutation invariance and equivariance, respectively. For the latter case, specialised geometric GNN models have been proposed, such as SchNet Schütt et al. (2018), DimeNet(++) Gasteiger et al. (2020b;a), PaiNN Schütt et al. (2021), NequIP Batzner et al. (2022), or a transformer-based TorchMD-Net Thölke & De Fabritiis (2022b).

In this work we are primarily motivated by the PCQM4Mv2 dataset that is a part of the large-scale graph ML challenge (Hu et al., 2021), which contains uniquely large numbers of graphs, and has the particular characteristic that the molecular 3D structure is only provided for the training portion of the dataset, but not at test time. This has motivated methods specifically equipped to handle such a scenario: Noisy Nodes denoising autoencoding (Godwin et al., 2022) and pretraining (Zaidi et al., 2022), GEM-2 (Liu et al., 2022a), ViSNet (Wang et al., 2022b). Methods based on global self-attention (Vaswani et al., 2017) became particularly dominant after Graphormer (Ying et al., 2021a) won OGB-LSC 2021, which spurred development of transformer-based methods: SAN (Kreuzer et al., 2021), EGT (Hussain et al., 2022), GPS (Rampášek et al., 2022), TokenGT (Kim et al., 2022), or Transformer-M (Luo et al., 2022).

## 3 Preliminaries

Throughout the paper we use the following notation. Bold lowercase letters $\mathbf{v}$ are (row) vectors, bold uppercase letters $\mathbf{M}$ are matrices, with individual elements denoted by non-bold letters i.e. $v_k$ or $M_{pq}$. Blackboard bold lowercase letters $\mathbb{v}$ are categorical (integer-valued) vectors. In general, we denote by $\left[\mathbf{v}_k\right]_{k \in K}$ the vertical concatenation (stacking) of vectors $\mathbf{v}_k$. Vertical concatenation is also denoted by a semicolon, i.e. $\left[\mathbf{v}_1 \; ; \; ... \; ; \; \mathbf{v}_J\right] = \left[\mathbf{v}_j\right]_{j=1}^J = \left[\mathbf{v}_j \text{ for } j \in \{1, ..., J\}\right]$. Horizontal concatenation, which typically means concatenation along the feature dimension, is denoted by a vertical bar, i.e. $\left[\mathbf{v}_1 \mid \mathbf{v}_2\right]$.

A molecule is represented as a graph $\mathcal{G} = (\mathcal{V}, \mathcal{E})$ for nodes $\mathcal{V}$ and edges $\mathcal{E}$. In this representation, each node $i \in \mathcal{V}$ is an atom in the molecule and each edge $(u, v) \in \mathcal{E}$ is a chemical bond between two atoms. The number of atoms in the molecule is denoted by $N = |\mathcal{V}|$ and the number of edges is $M = |\mathcal{E}|$.

Each node and edge is associated with a list of categorical features $\mathbb{x}_i \in \mathbb{Z}^{D_{\text{atom}}}$ and $\mathbb{e}_{uv} \in \mathbb{Z}^{D_{\text{bond}}}$, respectively, for $D_{\text{atom}}$ atom features and $D_{\text{bond}}$ bond features. A further set of 3D atom positions $\mathbf{R} = \left[\mathbf{r}_1 \; ; \; ... \; ; \; \mathbf{r}_N\right] \in \mathbb{R}^{N \times 3}$, extracted from original DFT calculations, is provided for training data, but crucially not for validation and test data.

Our algorithm operates on edge, node, and global *features*. Node features in layer $\ell$ are denoted by $\mathbf{x}_i^\ell \in \mathbb{R}^{d_{\text{node}}}$, and are concatenated into the $N \times d_{\text{node}}$ matrix $\mathbf{X}^\ell = \left[\mathbf{x}_1^\ell \; ; \; ... \; ; \; \mathbf{x}_N^\ell\right]$. Edge features $\mathbf{e}_{uv}^\ell \in \mathbb{R}^{d_{\text{edge}}}$ are concatenated into the edge feature matrix $\mathbf{E}^\ell = \left[\mathbf{e}_{uv}^\ell \text{ for } (u, v) \in \mathcal{E}\right]$. Global features are defined per layer as $\mathbf{g}^\ell \in \mathbb{R}^{d_{\text{global}}}$.

We also define an *attention bias* matrix $\mathbf{B} \in \mathbb{R}^{N \times N}$, computed from the input graph topology and 3D atom positions, described later in section 4.2.

## 4 GPS++

Our GPS++ model closely follows the GPS framework set out by Rampášek et al. (2022). This work presents a flexible model structure for building hybrid MPNN/Transformer models for graph-structured input data. We build a specific implementation of GPS that focuses on maximising the benefit of the inductive biases of the graph structure and 3D positional information. We do this by building a large and expressive MPNN component and biasing our attention component with structural and positional information. We also allow global information to be propagated through two mechanisms, namely the global attention and by using a global feature in the MPNN.

As displayed in Figure 1, the main `GPS++` block (section 4.1) combines the benefits of both message passing and attention layers by running them in parallel before combining them with a simple summation and MLP; this layer is repeated 16 times. This main trunk of processing is preceded by an `Encoder` function responsible for encoding the input information into the latent space (section 4.2), and is followed by a simple `Decoder` function (section 5.2).

Feature engineering is also used to improve the representation of the atoms/bonds, to provide rich positional and structural features that increase expressivity, and to bias the attention weights with a distance embedding.

### 4.1 GPS++ Block

The `GPS++` block is defined as follows for layers $\ell > 0$ (see section 4.2 for the definitions of $\mathbf{X}^0, \mathbf{E}^0, \mathbf{g}^0, \mathbf{B}$):

$$\mathbf{X}^{\ell+1}, \mathbf{E}^{\ell+1}, \mathbf{g}^{\ell+1} = \texttt{GPS++}\left(\mathbf{X}^\ell, \mathbf{E}^\ell, \mathbf{g}^\ell, \mathbf{B}\right) \tag{1}$$

computed as

$$\mathbf{Y}^\ell, \; \mathbf{E}^{\ell+1}, \; \mathbf{g}^{\ell+1} = \texttt{MPNN}\left(\mathbf{X}^\ell, \mathbf{E}^\ell, \mathbf{g}^\ell\right) \tag{2}$$

$$\mathbf{Z}^\ell = \texttt{BiasedAttn}\left(\mathbf{X}^\ell, \mathbf{B}\right) \tag{3}$$

$$\forall i : \quad \mathbf{x}_i^{\ell+1} = \texttt{FFN}\left(\mathbf{y}_i^\ell + \mathbf{z}_i^\ell\right) \tag{4}$$

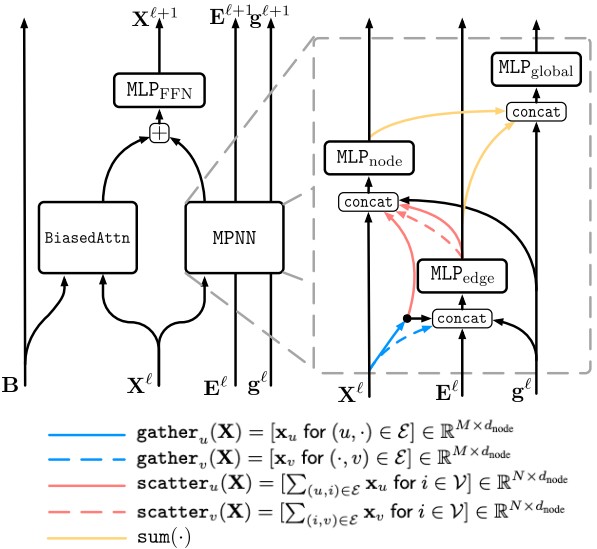

Figure 2: The main `GPS++` processing block (left) is composed of a local message passing `MPNN` module and a biased global attention `BiasedAttn` module. (right) A diagram of the used `MPNN` block. The `gather`, `scatter` and `sum` operations highlight changes in tensor shapes and are defined in equations 6 to 11.

Our `MPNN` module is a variation on the neural message passing module with edge and global features (Gilmer et al., 2017; Battaglia et al., 2018; Bronstein et al., 2021). We choose this form to maximise the expressivity of the model (Veličković, 2023) with the expectation that overfitting will be less of an issue with PCQM4Mv2, compared to other molecular datasets, due to its size. The essential components (excluding dropout and layer norm) of the `MPNN` module are defined as follows (see Figure 2 for a graphical representation, and Appendix A.1 for the exact formulation):

$$\mathbf{Y}^\ell, \ \mathbf{E}^{\ell+1}, \ \mathbf{g}^{\ell+1} = \texttt{MPNN}\left(\mathbf{X}^\ell, \mathbf{E}^\ell, \mathbf{g}^\ell\right) \tag{5}$$

computed as

$$\forall (u,v): \quad \bar{\mathbf{e}}_{uv}^\ell = \texttt{MLP}_{\text{edge}}\left(\left[\mathbf{x}_u^\ell \mid \mathbf{x}_v^\ell \mid \mathbf{e}_{uv}^\ell \mid \mathbf{g}^\ell\right]\right) \tag{6}$$

$$\forall i: \quad \bar{\mathbf{x}}_i^\ell = \texttt{MLP}_{\text{node}}\left(\left[\mathbf{x}_i^\ell \ \middle| \ \underbrace{\sum_{(u,i)\in\mathcal{E}} \bar{\mathbf{e}}_{ui}^\ell}_{\substack{\text{sender} \\ \text{messages}}} \ \middle| \ \underbrace{\sum_{(i,v)\in\mathcal{E}} \bar{\mathbf{e}}_{iv}^\ell}_{\substack{\text{receiver} \\ \text{messages}}} \ \middle| \ \underbrace{\sum_{(u,i)\in\mathcal{E}} \mathbf{x}_u^\ell}_{\substack{\text{adjacent} \\ \text{nodes}}} \ \middle| \ \mathbf{g}^\ell\right]\right) \tag{7}$$

$$\bar{\mathbf{g}}^\ell = \texttt{MLP}_{\text{global}}\left(\left[\mathbf{g}^\ell \ \middle| \ \sum_{j\in\mathcal{V}} \bar{\mathbf{x}}_j^\ell \ \middle| \ \sum_{(u,v)\in\mathcal{E}} \bar{\mathbf{e}}_{uv}^\ell\right]\right) \tag{8}$$

$$\forall i: \quad \mathbf{y}_i^\ell = \bar{\mathbf{x}}_i^\ell + \mathbf{x}_i^\ell \tag{9}$$

$$\forall (u,v): \quad \mathbf{e}_{uv}^{\ell+1} = \bar{\mathbf{e}}_{uv}^\ell + \mathbf{e}_{uv}^\ell \tag{10}$$

$$\mathbf{g}^{\ell+1} = \bar{\mathbf{g}}^\ell + \mathbf{g}^\ell \tag{11}$$

The three networks $\texttt{MLP}_\eta$ for $\eta \in \{\text{node}, \text{edge}, \text{global}\}$ each have two layers with `GELU` activation functions and an expanded intermediate hidden dimension of $4d_\eta$.

This message passing block is principally the most similar to Battaglia et al. (2018). However, we draw the reader's attention to a few areas that differ from common approaches. Firstly, we choose to decouple the three latent sizes, setting $d_{\text{node}} = 256$, $d_{\text{edge}} = 128$ and $d_{\text{global}} = 64$, because we found that increasing $d_{\text{edge}}$ and $d_{\text{global}}$ does not improve task performance. Secondly (and relatedly) we aggregate over the adjacent node

representations as well as the more complex edge feature messages in Equation 7, this is similar to running a simple Graph Convolutional Network (GCN) in parallel to the more complex message calculation. We do this to allow the higher-dimensional node features $\mathbf{x}_u^\ell$ to bypass the lower-dimensional messages $\bar{\mathbf{e}}_{uv}^\ell$, preventing potential compression. Finally, we also aggregate not only the *sender messages* from edges directed towards the node, but also the *receiver messages* computed on the edges directed away from the node (labelled in Equation 7). By concatenating these two terms rather than summing them we maintain directionality data in each edge, encouraging message information to flow bidirectionally but not requiring the MLP$_{\text{edge}}$ to learn directional invariance.

Our `BiasedAttn` module follows the form of the self-attention layer of Luo et al. (2022) where a standard self-attention block (Vaswani et al., 2017) is biased by a structural prior derived from the input data. In our work, the bias $\mathbf{B}$ is made up of two components, a shortest path distance (SPD) embedding and a 3D distance bias derived from the molecular conformations as described in section 4.2.

The `FFN` module takes a similar form to MLP$_{\text{node}}$ though with additional dropout terms (see Appendix A.1 for full details).

## 4.2 Input Feature Engineering

As described in section 3, the dataset samples include the graph structure $\mathcal{G}$, a set of categorical features for the atoms and bonds $\mathbb{x}_i$, $\mathbb{e}_{uv}$, and the 3D node positions $\mathbf{r}_i$. It has been shown that there are many benefits to augmenting the input data with additional structural, positional, and chemical information (Rampášek et al., 2022; Wang et al., 2022a; Dwivedi et al., 2022). Therefore, we combine several feature sources when computing the input to the first GPS++ layer. There are four feature tensors to initialise; node state, edge state, whole graph state and attention biases.

$$\mathbf{X}^{all} = \left[\mathbf{X}^{\text{atom}} \mid \mathbf{X}^{\text{LapVec}} \mid \mathbf{X}^{\text{LapVal}} \mid \mathbf{X}^{\text{RW}} \mid \mathbf{X}^{\text{Cent}} \mid \mathbf{X}^{\text{3D}}\right] \tag{12}$$

$$\mathbf{X}^0 = \texttt{Dense}(\mathbf{X}^{all}) \qquad \in \mathbb{R}^{N \times d_{\text{node}}} \tag{13}$$

$$\mathbf{E}^0 = \texttt{Dense}\left(\left[\mathbf{E}^{\text{bond}} \mid \mathbf{E}^{\text{3D}}\right]\right) \quad \in \mathbb{R}^{M \times d_{\text{edge}}} \tag{14}$$

$$\mathbf{g}^0 = \texttt{Embed}_{d_{\text{global}}}(0) \qquad \in \mathbb{R}^{d_{\text{global}}} \tag{15}$$

$$\mathbf{B} = \mathbf{B}^{\text{SPD}} + \mathbf{B}^{\text{3D}} \qquad \in \mathbb{R}^{N \times N} \tag{16}$$

Here the node features $\mathbf{X}^0$ are built from categorical atom features, graph Laplacian positional encodings (Kreuzer et al., 2021; Dwivedi & Bresson, 2020), random walk structural encodings (Dwivedi et al., 2022), local graph centrality encodings (Ying et al., 2021a; Shi et al., 2022) and a 3D centrality encoding (Luo et al., 2022). The edge features $\mathbf{E}^0$ are derived from categorical bond features and bond lengths, and the attention bias uses SPD and 3D distances. The global features are initialised with a learned constant embedding in latent space. These input features are further described in Appendix A.2.

**Chemical Features** The categorical features $\mathbb{x}$, $\mathbb{e}$ encapsulate the known chemical properties of the atoms and bonds, for example, the atomic number, the bond type or the number of attached hydrogens (which are not explicitly modelled as nodes), as well as graphical properties like node degree or whether an atom or bond is within a ring. The set that is used is not determined by the dataset and augmenting or modifying the chemical input features is a common strategy for improving results.

By default, the PCQM4Mv2 dataset uses a set of 9 atom and 3 bond features (described in Table A.1). There is, however, a wide range of chemical features that can be extracted from the periodic table or using tools like RDKit. Ying et al. (2021b) have shown that extracting additional atom level properties can be beneficial when trying to predict the HOMO-LUMO energy gap, defining a total of 28 atom and 5 bond features. We explore the impact of a number of additional node and edge features and sweep a wide range of possible combinations.

In particular, we expand on the set defined by Ying et al. (2021b) with three additional atom features derived from the periodic table, the atom group (column), period (row) and element type (often shown by colour). These are intended to generalise to unseen atom types better than a single integer atomic number feature, and we found that them to be particularly beneficial (shown later in ablation Table 5).

We also found that in many cases *removing* features was beneficial, for example, we found that generally our models performed better when excluding information about chiral tag and replacing it with chiral centers. We further observe that our best feature combinations all consist of only 8 node features, where the majority of the input features stay consistent between the sets. We show the three best feature sets found in Table A.1 and use *Set 1* for all experiments unless otherwise stated (i.e., during ablations).

**Comparison to GPS**   The GPS++ model is a specific instantiation of the GPS framework, and therefore has several similarities with the reference GPS model implemented by Rampášek et al. (2022). The high level architecture of each layer remains the same, using an MPNN module and self-attention module in parallel before combining their node updates with an MLP. However, the design of these two modules is quite different in our implementation. GPS++ uses a bespoke MPNN instead of the GatedGCN (Bresson & Laurent, 2018) employed by GPS. Our MPNN is most similar to Battaglia et al. (2018) but with some key differences described in section 4.1 such as GCN-like adjacent node aggregation and reverse message passing. For the self-attention module, where GPS uses unbiased attention like Vaswani et al. (2017), GPS++ implements learned bias terms $\mathbf{B}^{\text{SPD}}$ and $\mathbf{B}^{\text{3D}}$ to incorporate the graph structure. Considering the input features of each model, GPS++ and GPS share the use of the Graph Laplacian and Random walk positional/structural encodings $\mathbf{X}^{\text{LapVec}}$, $\mathbf{X}^{\text{LapVal}}$ and $\mathbf{X}^{\text{RW}}$, whilst GPS++ additionally uses node centrality encoding $\mathbf{X}^{\text{Cent}}$ and the spatial node and edge features $\mathbf{X}^{\text{3D}}$ and $\mathbf{E}^{\text{3D}}$. Moreover, the atomic features $\mathbf{X}^{\text{atom}}$ in GPS++ are a refinement of the defaults used by GPS, including the addition of the novel atomic Group, Period and Element Type features. Finally, while GPS trains by minimising the error on only the HOMO-LUMO prediction task, GPS++ utilises the following extra regularisation techniques in order to effectively scale parameters on finite data: a node/edge feature denoising task (Godwin et al., 2022), a 3D spatial denoising task (Luo et al., 2022), stochastic depth (Huang et al., 2016) and pervasive element-wise dropout. The additional training tasks are described further in section 5.

## 5   Experimental Setup

### 5.1   Dataset

The PCQM4Mv2 dataset (Hu et al., 2021) consists of 3.7M molecules defined by their SMILES strings. Each molecule has on average 14 atoms and 15 chemical bonds. However, as the bonds are undirected in nature and graph neural networks act on directed edges, two bidirectional edges are used to represent each chemical bond.

The 3.7M molecules are separated into standardised sets, namely into `training` (90%), `validation` (2%), `test-dev` (4%) and `test-challenge` (4%) sets using a scaffold split where the HOMO-LUMO gap targets are only publicly available for the `training` and `validation` splits. PCQM4Mv2 also provides a conformation for each molecule in the training split, i.e., a position in 3D space for each atom such that each molecular graph is in a relaxed low-energy state. Crucially, the validation and test sets have no such 3D information provided.

### 5.2   Model Training

**Training Configuration**   Our model training setup uses the Adam optimiser (Kingma & Ba, 2015) with a gradient clipping value of 5, a peak learning rate of 4e-4 and the model is trained for a total of 450 epochs. We used a learning rate warmup period of 10 epochs followed by a linear decay schedule.

**Decoder and Loss**   The final model prediction is formed by global sum-pooling of all node representations and then passing it through a 2-layer MLP. The regression loss is the mean absolute error (L1 loss) between a scalar prediction and the ground truth HOMO-LUMO gap value.

**Noisy Nodes/Edges**   Noisy nodes (Godwin et al., 2022; Zaidi et al., 2022) has previously been shown to be beneficial for molecular GNNs including on the PCQM4M dataset. The method adds noise to the input data then tries to reconstruct the uncorrupted data in an auxiliary task. Its benefits are expected to be two-fold: it adds regularisation by inducing some noise on the input, but also combats oversmoothing by forcing the node-level information to remain discriminative throughout the model. This has been shown to be particularly

beneficial when training deep GNNs (Godwin et al., 2022). We follow the method of Godwin et al. (2022) that applies noise to the categorical node features by randomly choosing a different category with probability $p_{\text{corrupt}}$, and we further extend this to the categorical edge features. A simple categorical cross-entropy loss is then used to reconstruct the uncorrupted features at the output. We set $p_{\text{corrupt}} = 0.01$ and weight the cross-entropy losses such that they have a ratio 1:1.2:1.2 for losses `HOMO-LUMO:NoisyNodes:NoisyEdges`. The impact of Noisy Nodes/Edges can be seen in Table 4 of the ablation study.

**Grouped Input Masking**   The 3D positional features $\mathbf{R}$ are only defined for the training data. We must therefore make use of them in training without requiring them for validation/test. We found that the method proposed by Luo et al. (2022) achieved the most favourable results so we adopt a variation hereon referred to as *grouped input masking.*

Atom distances calculated from 3D positions are embedded into vector space $\mathbb{R}^K$ via $K = 128$ Gaussian kernel functions. We then process these distance embeddings in three ways to produce attention biases ($\mathbf{B}^{\text{3D}}$), node features ($\mathbf{X}^{\text{3D}}$) and edge features ($\mathbf{E}^{\text{3D}}$) (exact formulations can be found in Appendix A.2).

This method stochastically masks out any features derived from the 3D positional features $\mathbf{R}$ to build robustness to their absence. Specifically, this is done by defining two input sets to be masked:

$$\mathcal{X}^{\text{Spatial}} = \{\mathbf{X}^{\text{3D}}, \mathbf{E}^{\text{3D}}, \mathbf{B}^{\text{3D}}\}, \quad \mathcal{X}^{\text{Topological}} = \{\mathbf{B}^{\text{SPD}}\}$$

and three potential masking groups: 1. Mask $\mathcal{X}^{\text{Spatial}}$, 2. Mask $\mathcal{X}^{\text{Topological}}$, and 3. No masking. These masking groups are then sampled randomly throughout training with ratio 1:3:1. If 3D positions are not defined, e.g. in validation/test, masking group 1 is always used.

**3D Denoising**   Alongside *grouped input masking*, Luo et al. (2022) also add noise to the 3D atom positions $\mathbf{R}$ during training before computing any derivative features (e.g., 3D Bias, 3D Centrality), and predict the atom-wise noise as an auxiliary self-supervised training task. We closely follow their implementation for GPS++, adding an SE(3)-equivariant self-attention layer as a noise prediction head, which encourages the model to preserve 3D information until the final layer as well as further develop spatial reasoning. See Appendix section A.2 for full details.

## 6   Results

**Model Accuracy**   In Table 2, we compare the single model performance of GPS++ with results from the literature, in particular, we pay close attention to the Transformer-M (Luo et al., 2022) model as it has the best results for a transformer, but also because we use their input masking and denoising method for incorporating 3D features; this allows us to gain some tangible insights into the value of hybrid/MPNN approaches vs. transformers.

Comparing the best GPS++ result with prior work, we set a new state of the art MAE score on the PCQM4Mv2 validation set, outperforming not only the parametrically comparable Transformer-M Medium but also the much larger Transformer-M Large. While it would be tempting to assume that the Multi-Headed Self Attention (MHSA) component of our model is the most significant due to the prevalence of transformers among the other top results, we find that we can maintain competitive accuracy with no attention at all, falling back to a pure MPNN, passing messages only between bonded atoms (nodes). Furthermore, we show that this MPNN model is more accurate in the absence of 3D features, even beating the hybrid GPS++ in this setting. We believe this shows a strong reliance of the transformer's global attention mechanism on 3D positional information to learn atom relationships and, in contrast, the power of encoding spatial priors into the model using message passing along molecular bonds in the MPNN. Further, by halving the number of GPS++ layers from 16 to 8 to reach parametric parity, Table 2 shows that the MPNN-only GPS++ model compares favourably to the original GPS implementation. We investigate the impact of other components of the model more in section 7.

**Model Throughput**   Molecules in PCQM4Mv2 have an average of 14.3 nodes (atoms) per graph, and our full GPS++ model trains at 17,500 graphs per second on 16 IPUs. This means each epoch completes in

Table 2: Comparison of single model performance on PCQM4Mv2 dataset. Note Transformer-M and Global-ViSNet are concurrent work.

| Model | # Param. | Model Type | Valid MAE (meV) ↓ |
|---|---|---|---|
| *without 3D Positional Information* | | | |
| GCN-virtual (Hu et al., 2021) | 4.9M | MPNN | 115.3 |
| GIN-virtual (Hu et al., 2021) | 6.7M | MPNN | 108.3 |
| GRPE (Park et al., 2022) | 46.2M | Transformer | 89.0 |
| Transformer-M [Medium, No 3D Positions] (Luo et al., 2022) | 47.1M | Transformer | 87.8 |
| EGT (Hussain et al., 2022) | 89.3M | Transformer | 86.9 |
| Graphormer (Shi et al., 2022) | 48.3M | Transformer | 86.4 |
| GPS (Rampášek et al., 2022) | 19.4M | Hybrid | 85.8 |
| GPS++ [8 Layers, No 3D Positions] | 22.4M | Hybrid | 83.6 |
| GPS++ [No 3D Positions] | 44.2M | Hybrid | 82.6 |
| GPS++ [8 Layers, MPNN only, No 3D Positions] | 20.3M | MPNN | 82.2 |
| GPS++ [MPNN only, No 3D Positions] | 40.0M | MPNN | **81.8** |
| *with 3D Positional Information* | | | |
| GEM-2 (Liu et al., 2022a) | 32.1M | Transformer | 79.3 |
| GPS++ [8 Layers, MPNN only] | 22.5M | MPNN | 79.3 |
| Transformer-M [Medium] (Luo et al., 2022) | 47.1M | Transformer | 78.7 |
| GPS++ [8 Layers] | 22.7M | Hybrid | 78.6 |
| Global-ViSNet (Wang et al., 2022b) | 78.5M | Transformer | 78.4 |
| Transformer-M [Large] (Luo et al., 2022) | 69.0M | Transformer | 77.2 |
| GPS++ [MPNN only] | 40.3M | MPNN | 77.2 |
| GPS++ | 44.5M | Hybrid | **76.6** |

3 minutes and a full 450 epoch training run takes less than 24 hours. The MPNN-only version of GPS++ (i.e. disabling the MHSA module) has almost double the throughput at 33,000 graphs per second, partially due to the reduced compute costs but also as a result of increasing the micro batch size thanks to reduced memory pressure. Figure 3 compares the models for a range of training epochs, and shows the MPNN-only version can outperform prior results from GEM-2 and Global-ViSNet with only 5-6 hours of training, and match the prior SOTA Transformer-M after 13 hours. High throughput allows for rapid iteration and extensive ablation studies (see section 7). Full details of the hardware used and acceleration methods employed can be found in Appendix B.

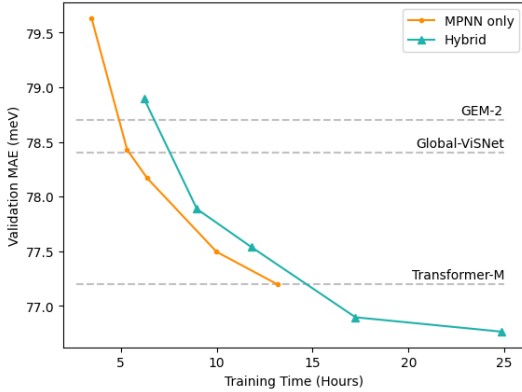

Figure 3: Training time versus validation error for the full hybrid GPS++ model versus the MPNN-only model. Comparable training times for GEM-2, Global-ViSNet and Transformer-M were unavailable at the time of writing.

**Fine-tuning Accuracy**   In order to test GPS++ in the presence of 3D positional data during test time and to investigate the generalisability of the model, we fine-tune GPS++ on 8 different tasks from the the quantum chemistry benchmark QM9 (Ruddigkeit et al., 2012; Ramakrishnan et al., 2014). The QM9 dataset comprises 134k small organic molecules in equilibrium state which contain up to 9 heavy atoms (C, O, N, F) and, unlike in the PCQ dataset, 3D positional information is available at test time. Though the benchmark provides 12

labels for each molecule, we found that 4 very closely related energy labels $U_0, U, H$ and $G$ were particularly sensitive to hyperparameters, random seeds and label normalisation choices, so we have omitted these labels to reduce the time and compute costs of the generalisation study. QM9 does not provide a standardised dataset split, we therefore follow several previous works (Luo et al., 2022; Thölke & De Fabritiis, 2022a) and randomly select 10,000 molecules for validation and 10,831 for testing; all remaining molecules are used during training. Due to the small size of this dataset and the high variance in test accuracy observed for different train/test splits, we fine-tune 3 baseline GPS++ checkpoints on 5 random split seeds for each task and report both the mean and standard deviation across all 15 runs. We disable grouped input masking during fine-tuning, to maximise the benefit of QM9's always-present 3D features. For each label we performed a hyperparameter sweep, settling on learning rates in the range $[0.0004, 0.001]$, epochs in the range $[1000, 2000]$, 3D denoising loss weighting in the range $[0.00001, 0.75]$, and in some cases removing some dropout operations. In particular, we found that the HOMO, LUMO and Gap labels benefited from a much higher 3D denoising loss weight than pretraining (0.75 vs 0.1). By contrast, for the label $R^2$ we found the best results by setting the 3D denoising weight to 0.00001 and disabled dropout in the node MLP as well as stochastic depth, suggesting the signal-to-noise ratio in this label is much lower.

The results are shown in Table 3. Despite differences in the types of molecules, the DFT configurations used to generate the equilibrium states and the availability of 3D positions at test time, GPS++ generalises very well to this new dataset for tasks like HOMO, LUMO and Gap which are closely related to the PCQM4Mv2 pretraining task. The performance of GPS++ on less closely related labels generally falls behind the state of the art, the weakest being ZPVE and $R^2$, though in each case the results are well within the distribution of reasonable results from prior work. This bias of the model to a particular kind of task appears to be the norm throughout the QM9 results; each model that sets the state of the art in one label also exhibits particularly weak performance in one or more other labels. Beyond its current strengths, it may be possible to improve GPS++'s weaker scores by implementing task-specific prediction heads. For example, Thölke & De Fabritiis (2022a) add a rotationally-equivariant output layer just when predicting the Electronic Spatial Extent ($R^2$), as this is a highly spatial task.

Table 3: QM9 fine-tuning results. Test MAE and STD reported over 5 test split seeds and 3 pre-trained GPS++ checkpoints (15 runs total). The **first** and **second** best results for each label are highlighted.

| Model | Pretrained | Homo meV | Lumo meV | Gap meV | $C_V$ cal/mol K | $\mu$ D | ZPVE meV | $R^2$ $\alpha_0^2$ | $\alpha$ $\alpha_0^3$ |
|---|---|---|---|---|---|---|---|---|---|
| Schnet (Schütt et al., 2018) | · | 41 | 34 | 63 | 0.033 | 0.033 | 1.70 | 0.073 | 0.24 |
| MGCN (Lu et al., 2019) | · | 42 | 57 | 64 | 0.038 | 0.056 | **1.12** | 0.11 | **0.030** |
| LieConv (Finzi et al., 2020) | · | 30 | 25 | 49 | 0.038 | 0.032 | 2.28 | 0.80 | 0.084 |
| SE(3)-Transformer (Fuchs et al., 2020) | · | 35 | 33 | 53 | 0.054 | 0.051 | - | - | 0.14 |
| DimeNet++ (Gasteiger et al., 2020a) | · | 24.6 | 19.5 | 32.6 | 0.023 | 0.030 | 1.21 | 0.33 | 0.044 |
| GEM (Fang et al., 2021) | ✓ | 33.8 | 27.7 | 52.1 | 0.035 | 0.034 | 1.73 | 0.089 | 0.081 |
| PaiNN (Schütt et al., 2021) | · | 27.6 | 20.4 | 45.7 | 0.024 | 0.012 | 1.28 | **0.066** | 0.045 |
| LieTF (Hutchinson et al., 2021) | · | 33 | 27 | 51 | 0.035 | 0.041 | 2.10 | 0.45 | 0.082 |
| TorchMD-Net (Thölke & De Fabritiis, 2022b) | · | 20.3 | 17.5 | 36.1 | 0.026 | **0.011** | 1.84 | **0.033** | 0.059 |
| EGNN (Satorras et al., 2021) | · | 29 | 25 | 48 | 0.031 | 0.029 | 1.55 | 0.11 | 0.071 |
| SphereNet (Liu et al., 2022b) | · | 22.8 | 18.9 | 31.1 | **0.022** | 0.026 | **1.12** | 0.29 | 0.046 |
| SEGNN (Brandstetter et al., 2022) | · | 24 | 21 | 42 | 0.031 | 0.023 | 1.62 | 0.66 | 0.060 |
| EQGAT (Le et al., 2022) | · | 20 | 16 | 32 | 0.024 | **0.011** | 2.00 | 0.38 | 0.053 |
| 3D Infomax (Stärk et al., 2022) | ✓ | 29.8 | 25.7 | 48.8 | 0.033 | 0.034 | 1.67 | 0.12 | 0.075 |
| 3D-MGP (Jiao et al., 2022) | ✓ | 21.3 | 18.2 | 37.1 | 0.026 | 0.020 | 1.38 | 0.092 | 0.057 |
| NoisyNode (Godwin et al., 2022) | · | 20.4 | 18.6 | 28.6 | 0.025 | 0.025 | 1.16 | 0.70 | 0.052 |
| Transformer-M (Luo et al., 2022) | ✓ | 17.5 | 16.2 | 27.4 | **0.022** | 0.037 | 1.18 | 0.075 | 0.041 |
| GNS-TAT+NN (Zaidi et al., 2022) | ✓ | **14.9** | **14.7** | **22.0** | **0.020** | 0.016 | **1.02** | 0.44 | **0.040** |
| GPS++ | ✓ | **16.1** ± 0.3 | **14.6** ± 0.4 | **26.2** ± 0.3 | 0.028 ± 0.001 | 0.024 ± 0.001 | 1.85 ± 0.28 | 0.40 ± 0.03 | 0.062 ± 0.005 |

## 7 Ablation Study

The top-performing GPS++ model was attained empirically, resulting in a complex combination of input features, architectural choices and loss functions. In this section we assess the contribution of each feature to final task performance on PCQM4Mv2, and find that much of the performance can be retained by a simplified design. All results listed in the ablation study are obtained by training the final GPS++ model from scratch

for 200 epochs with one or more features disabled, averaging MAE over 5 runs. Average batch size is kept constant (926 nodes per batch) for all runs to keep them directly comparable.

**Node and Edge Input Features**   The PCQM4Mv2 dataset provides 9 chemical node and 3 edge features for each sample, and following prior work GPS++ incorporates additional features and preprocessing strategies. Table 4 isolates the contribution of each of these additions (excluding 3D features).

Table 4: Ablation of node and edge features.

| Removed Feature | Δ MAE (meV) | |
| --- | --- | --- |
| | **Valid** | **Train** |
| None (Baseline) | 77.2 | 53.6 |
| Random Walk Structural Encoding | **+ 1.2** | + 1.5 |
| Shortest Path Distance Bias | **+ 0.8** | + 2.6 |
| Laplacian Positional Encoding | **+ 0.3** | + 0.1 |
| Local Centrality Encoding | **+ 0.1** | 0.0 |
| Bond Lengths | **+ 0.1** | - 0.1 |
| Noisy 3D Positions | **+ 1.6** | + 3.6 |
| Noisy Nodes | **+ 0.6** | + 0.2 |
| Noisy Edges | **+ 0.1** | - 0.1 |

In line with previous work (Rampášek et al., 2022), the Random Walk Structural Encoding is the most impactful individual input feature, degrading validation performance by 1.2 meV upon removal, whilst the Graph Laplacian Positional Encodings (both eigenvectors and eigenvalues) have minimal benefit. The Shortest Path Distance Bias in the self-attention module makes a meaningful contribution, though the memory cost of storing all-to-all node distances is non-trivial. Other features, despite being beneficial when they were added at intermediate steps in the development of the model, are shown to have become redundant in the final GPS++ composition: the Graphormer-style Local Centrality Encoding (i.e. embedding the degree of the node) can be derived from the Random Walk features, and the use of Bond Lengths as an edge feature in the MPNN appears to be unnecessary in the presence of more comprehensive 3D features elsewhere in the network. Table 4 also shows that whilst the regularisation effect of the noisy nodes loss is beneficial, the contribution of noisy edges is negligible. This may be due to the simplicity (and hence ease of denoising) of the limited edge features in PCQM4Mv2.

Table 5: Choices of chemical features.

| Feature Set | Atom Group, Period & Type | Set Size | Δ MAE (meV) | |
| --- | --- | --- | --- | --- |
| | | | **Valid** | **Train** |
| Set 1 (Baseline) | ✓ | 14 | 77.2 | 53.6 |
| Set 2 | ✓ | 14 | **+ 0.2** | 0.0 |
| Set 3 | ✓ | 14 | **+ 0.2** | + 0.3 |
| Original | ✓ | 15 | **+ 0.2** | + 0.3 |
| Original | · | 12 | **+ 0.5** | + 0.7 |
| Set 1 | · | 11 | **+ 0.6** | + 0.4 |
| Ying21 | ✓ | 36 | **+ 4.1** | - 3.1 |
| Ying21 | · | 33 | **+ 4.4** | - 3.0 |

Table 5 compares the possible choices of chemical input features described in section 4.2: *Original* is the set of features included in PCQM4Mv2; *Ying21* refers to all 28 node features and 5 edge features in the PCQ superset defined by Ying et al. (2021b); *Set 1-3* are performant subsets of 11 node features and 3 edge features selected from all of the above, defined fully in Table A.1 and obtained via a combinatorial search; *Atom Group, Period & Type* refers to our contribution of 3 additional atom features relating to an atom's position in the periodic table, intended to be more generalised than the atomic number. The results clearly show that training on all 36 available input features causes significant overfitting and degrades task performance by introducing noise. Moreover, the new *Atom Group, Period & Type* features consistently provide meaningfully richer atom representations than the atomic number feature which is present in every feature set. We believe future work on similar datasets should consider incorporating this simple addition.

**3D Input Features and Self-Attention**  The 3D node position information given for the training set in PCQM4Mv2 is provided as input to the final model in three forms: the Bond Length edge features in the MPNN, the all-to-all 3D Bias map added to the self-attention module, and the node-wise 3D Centrality Encoding added in the node encoder. The Bond Lengths are shown to have minimal impact in Table 4, so Table 6 explores the 3D Bias and Centrality as well as how they interact with the MHSA module. In rows 1-4 we see that including at least one of the pair is critical to the final model's performance, but neither makes the other redundant. The 3D Bias is higher fidelity than the 3D Centrality (i.e, an all-to-all distance map versus a node-wise distance sum) and is the stronger of the pair, but the effectiveness of the simpler 3D Centrality should be noted since it does not require the use of custom biased self-attention layers. Additionally, we found that training stability of the MHSA module suffered in the absence of either 3D feature, requiring us to halve the learning rate to 2e-4 for convergence unless the MHSA module was disabled.

Table 6: Ablation of self-attention and 3D features.

| # | Feature | | | $\Delta$ **MAE (meV)** | | |
| | **MHSA** | **3D Bias** | **3D Centrality** | **Valid** | **Train** | |
|---|---|---|---|---|---|---|
| 1 | ✓ | ✓ | ✓ | 77.2 | 53.6 | |
| 2 | ✓ | ✓ | · | + 0.6 | - 2.3 | ∗ |
| 3 | ✓ | · | ✓ | + 1.1 | - 3.6 | ∗ |
| 4 | ✓ | · | · | + 5.1 | - 10.0 | ∗ |
| 5 | · | · | · | + 4.3 | - 16.4 | |
| 6 | · | · | ✓ | + 0.9 | - 1.5 | |

∗ trained with half learning rate for stability

Interestingly, when the 3D Bias term is not used, Table 6 rows 4-5 show that it is preferable to *remove* the MHSA. This may imply that the Shortest Path Distance Bias term (the other bias in the MHSA, which attempts to provide a non-3D global graph positional encoding) is insufficient to modulate attention between atoms, and that instead spatial relationships are key for this quantum chemistry task. For example, when a molecule is folded over and two atoms may be very close in 3D space but far apart in the graph topology.

**Model Architecture**  GPS++ uses a large MPNN module comprising 70% of the model parameters and using extensively tuned hyper-parameters. Table 7 ablates the components of the model network architecture, and shows that the MPNN module is unsurprisingly the single most important component for task performance. Outside of the MPNN, the MHSA (with accompanying attention biases) and FFN contribute approximately equally to the final performance, though they have opposite impacts on training MAE. Within the MPNN, the largest task impact arises from removing the edge features and edge MLP from all layers ($\mathbf{e}_{uv}$ and $\mathsf{MLP}_{\text{edge}}$ in Equation 6), falling back on simple GCN-style message passing that concatenates neighbouring node features when computing messages. Table 7 also shows that the use of Adjacent Node Feature Aggregation (i.e., the sum of adjacent node features $\mathbf{x}_u^\ell$ in Equation 7) allows large node features of size 256 to bypass compression into edge messages of size 128, affording a small but meaningful MAE improvement. It is likely due to this feature that we have not found a benefit to increasing the edge latent size to match the node size. Both the use of Global Features and Sender Message Aggregation within the MPNN are shown to be of comparable importance to the MHSA and FFN modules. Global Features, denoted as $\mathbf{g}^\ell$ in Equation 6 and 7, allow the MPNN some degree of global reasoning, whilst Sender Message Aggregation (the sum of $\bar{\mathbf{e}}_{iv}^\ell$ in Equation 7) uses outgoing messages in addition to the usual incoming messages to update a node's features. There is some overlap in purpose of the Global Features and the MHSA module, so the table also shows the impact of removing both at once. The resulting performance degradation is significantly greater than the sum of their individual impacts, which may imply each feature is able to partially compensate for the loss of the other in the individual ablations.

Table 7: Ablation of network architecture features.

| | $\Delta$ MAE (meV) | | |
| --- | --- | --- | --- |
| **Removed Feature** | **Valid** | **Train** | **Params ($\Delta$)** |
| None (Baseline) | 77.2 | 53.6 | 44M (  0%) |
| MPNN | **+ 7.7** | + 10.9 | 14M (-70%) |
|   Edge Features | **+ 4.2** | + 3.7 | 33M (-26%) |
|   Global Features | **+ 1.2** | 0.0 | 40M ( -8%) |
|   Sender Message Aggregation | **+ 0.7** | + 0.4 | 38M (-14%) |
|   Adjacent Node Aggregation | **+ 0.4** | + 1.3 | 36M (-19%) |
| MHSA | **+ 0.9** | - 1.5 | 40M (-10%) |
| FFN | **+ 1.0** | + 1.3 | 36M (-19%) |
| MHSA and Global Features | **+ 3.4** | - 0.2 | 37M (-18%) |

## 8  Discussion

In this work, we define GPS++, a hybrid MPNN/Transformer, optimised for the PCQM4Mv2 molecular property prediction task (Hu et al., 2021). Our model builds on previous work (Rampášek et al., 2022; Luo et al., 2022; Godwin et al., 2022) with a particular focus on building a powerful and expressive message passing component comprising 70% of model parameters. We showed that our GPS++ model has state-of-the-art performance on this uniquely large-scale dataset, and that despite recent trends towards using only graph transformers in molecular property prediction, GPS++ retains almost all of its performance as a pure MPNN with the global attention module ablated. Finally, we consider the case where 3D positions are not available and show that our GPS++ models are significantly better than transformers, where a significant drop-off is observed across all prior work. We also found that our MPNN-only model performs better than our hybrid under these circumstances; this indicates a strong reliance between effective attention and the availability of 3D positional information.

Whilst these results are interesting for problems with small molecules, testing on much larger molecules, for example, peptides, proteins, or RNAs, which can have hundreds or thousands of atoms, is a tougher challenge. Under these conditions, the linear complexity of MPNNs with graph size suggests some computational benefits over global attention, however, it is still to be determined if the downsides of issues like underreaching outweigh these benefits. Nevertheless, we believe that our results highlight the strength of MPNNs in this domain and hope that it inspires a revival of message passing and hybrid models for molecular property prediction when dealing with large datasets.

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

## A  Detailed Model Description

### A.1  GPS++ Block Complete

The GPS++ block is defined as follows for layers $\ell > 0$ (see section A.2 for the definitions of $\mathbf{X}^0, \mathbf{E}^0, \mathbf{g}^0$).

$$\mathbf{X}^{\ell+1}, \mathbf{E}^{\ell+1}, \mathbf{g}^{\ell+1} = \text{GPS++} \left( \mathbf{X}^\ell, \mathbf{E}^\ell, \mathbf{g}^\ell, \mathbf{B} \right) \tag{17}$$

computed as

$$\mathbf{Y}^\ell, \ \mathbf{E}^{\ell+1}, \ \mathbf{g}^{\ell+1} = \text{MPNN} \left( \mathbf{X}^\ell, \mathbf{E}^\ell, \mathbf{g}^\ell \right) \tag{18}$$

$$\mathbf{Z}^\ell = \text{BiasedAttn} \left( \mathbf{X}^\ell, \mathbf{B} \right) \tag{19}$$

$$\forall i : \quad \mathbf{x}_i^{\ell+1} = \text{FFN} \left( \mathbf{y}_i^\ell + \mathbf{z}_i^\ell \right) \tag{20}$$

**The MPNN module** A simplified version of the MPNN module is presented in section 4.1. The full description is as follows:

$$\mathbf{Y}^\ell, \ \mathbf{E}^{\ell+1}, \ \mathbf{g}^{\ell+1} = \texttt{MPNN}\left(\mathbf{X}^\ell, \mathbf{E}^\ell, \mathbf{g}^\ell\right) \tag{21}$$

computed as

$$\forall (u,v): \quad \mathbf{c}_{uv}^\ell = \left[\mathbf{x}_u^\ell \mid \mathbf{x}_v^\ell \mid \mathbf{e}_{uv}^\ell \mid \mathbf{g}^\ell\right] \tag{22}$$

$$\forall (u,v): \quad \bar{\mathbf{e}}_{uv}^\ell = \texttt{Dropout}_{0.0035}\left(\texttt{MLP}_{\text{edge}}\left(\mathbf{c}_{uv}^\ell\right)\right) \tag{23}$$

$$\forall i: \quad \bar{\mathbf{x}}_i^\ell = \texttt{MLP}_{\text{node}}\left(\left[\mathbf{x}_i^\ell \; \middle| \; \underbrace{\sum_{(u,i)\in\mathcal{E}} \bar{\mathbf{e}}_{ui}^\ell}_{\substack{\text{sender}\\\text{messages}}} \; \middle| \; \underbrace{\sum_{(i,v)\in\mathcal{E}} \bar{\mathbf{e}}_{iv}^\ell}_{\substack{\text{receiver}\\\text{messages}}} \; \middle| \; \underbrace{\sum_{(u,i)\in\mathcal{E}} \mathbf{x}_u^\ell}_{\substack{\text{adjacent}\\\text{nodes}}} \; \middle| \; \mathbf{g}^\ell\right]\right) \tag{24}$$

$$\bar{\mathbf{g}}^\ell = \texttt{MLP}_{\text{global}}\left(\left[\mathbf{g}^\ell \; \middle| \; \sum_{j\in\mathcal{V}} \bar{\mathbf{x}}_j^\ell \; \middle| \; \sum_{(u,v)\in\mathcal{E}} \bar{\mathbf{e}}_{uv}^\ell\right]\right) \tag{25}$$

$$\forall i: \quad \mathbf{y}_i^\ell = \texttt{LayerNorm}(\texttt{Dropout}_{0.3}(\bar{\mathbf{x}}_i^\ell) + \mathbf{x}_i^\ell) \tag{26}$$

$$\forall (u,v): \quad \mathbf{e}_{uv}^{\ell+1} = \bar{\mathbf{e}}_{uv}^\ell + \mathbf{e}_{uv}^\ell \tag{27}$$

$$\mathbf{g}^{\ell+1} = \texttt{Dropout}_{0.35}(\bar{\mathbf{g}}^\ell) + \mathbf{g}^\ell \tag{28}$$

where $\texttt{Dropout}_p$ (Srivastava et al., 2014) masks by zero each element with probability $p$ and $\texttt{LayerNorm}$ follows the normalisation procedure of Ba et al. (2016). The three networks $\texttt{MLP}_\eta$ for $\eta \in \{\text{node}, \text{edge}, \text{global}\}$ each have two layers and are defined by:

$$\mathbf{y} = \texttt{MLP}_\eta(\mathbf{x}) \tag{29}$$

computed as

$$\bar{\mathbf{x}} = \texttt{GELU}(\texttt{Dense}(\mathbf{x})) \qquad \in \mathbb{R}^{4d_\eta} \tag{30}$$

$$\mathbf{y} = \texttt{Dense}(\texttt{LayerNorm}(\bar{\mathbf{x}})) \qquad \in \mathbb{R}^{d_\eta} \tag{31}$$

where GELU is from Hendrycks & Gimpel (2016).

**The BiasedAttn module** Our BiasedAttn module follows the form of the self-attention layer in Luo et al. (2022) where a standard self-attention block (Vaswani et al., 2017) is biased by a structural prior derived from the input graph. In our work the bias B is made up of two components, a Shortest Path Distance embedding and a 3D Distance Bias derived from the molecular conformations as described in section A.2. Single-head biased attention is defined by:

$$\mathbf{Z} = \texttt{BiasedAttn}(\mathbf{X}, \mathbf{B}) \tag{32}$$

computed as

$$\mathbf{A} = \frac{(\mathbf{X}\mathbf{W}_Q)(\mathbf{X}\mathbf{W}_K)^\top}{\sqrt{d_{\text{node}}}} + \mathbf{B} \qquad \in \mathbb{R}^{N\times N} \tag{33}$$

$$\bar{\mathbf{A}} = \texttt{Dropout}_{0.3}\left(\texttt{Softmax}\left(\mathbf{A}\right)\right)(\mathbf{X}\mathbf{W}_V) \qquad \in \mathbb{R}^{N\times d_{\text{node}}} \tag{34}$$

$$\mathbf{Z} = \texttt{GraphDropout}_{\frac{\ell}{L}0.3}\left(\bar{\mathbf{A}}\right) + \mathbf{X} \qquad \in \mathbb{R}^{N\times d_{\text{node}}} \tag{35}$$

for learnable weight matrices $\mathbf{W}_Q, \mathbf{W}_K, \mathbf{W}_V \in \mathbb{R}^{d_{\text{node}}\times d_{\text{node}}}$ and output in $\mathbb{R}^{N\times d_{\text{node}}}$, though in practice we use 32 attention heads which are mixed before GraphDropout using an affine projection $\mathbf{W}_P \in \mathbb{R}^{d_{\text{node}}\times d_{\text{node}}}$.

**The FFN module**  Finally, the feed-forward network module takes the form:

$$\mathbf{y} = \text{FFN}(\mathbf{x}) \tag{36}$$

computed as

$$\bar{\mathbf{x}} = \text{Dropout}_p(\text{GELU}(\text{Dense}(\mathbf{x}))) \qquad \in \mathbb{R}^{4d_{\text{node}}} \tag{37}$$

$$\mathbf{y} = \text{GraphDropout}_{\frac{\ell}{L}0.3}(\text{Dense}(\bar{\mathbf{x}})) + \mathbf{x} \quad \in \mathbb{R}^{d_{\text{node}}} \tag{38}$$

Unless otherwise stated, the dropout probability $p = 0$. GraphDropout, also known as Stochastic Depth or LayerDrop (Huang et al., 2016), masks whole graphs together rather than individual nodes or features, relying on skip connections to propagate activatitions.

## A.2  Input Feature Engineering

As described in section 3, the dataset samples include the graph structure $\mathcal{G}$, a set of categorical features for the atoms and bonds $\mathbb{x}_i$, $\mathbb{e}_{uv}$, and the 3D node positions $\mathbf{r}_i$. It has been shown that there are many benefits to augmenting the input data with additional structural, positional, and chemical information (Rampášek et al., 2022; Wang et al., 2022a; Dwivedi et al., 2022). Therefore, we combine several feature sources when computing the input to the first GPS++ layer. There are four feature tensors to initialise; node state, edge state, whole graph state and attention biases.

$$\mathbf{X}^{all} = \left[\mathbf{X}^{\text{atom}} \mid \mathbf{X}^{\text{LapVec}} \mid \mathbf{X}^{\text{LapVal}} \mid \mathbf{X}^{\text{RW}} \mid \mathbf{X}^{\text{Cent}} \mid \mathbf{X}^{\text{3D}}\right]$$

$$\mathbf{X}^0 = \text{Dense}(\mathbf{X}^{all}) \qquad \in \mathbb{R}^{N \times d_{\text{node}}} \tag{39}$$

$$\mathbf{E}^0 = \text{Dense}(\left[\mathbf{E}^{\text{bond}} \mid \mathbf{E}^{\text{3D}}\right]) \quad \in \mathbb{R}^{M \times d_{\text{edge}}} \tag{40}$$

$$\mathbf{g}^0 = \text{Embed}_{d_{\text{global}}}(0) \qquad \in \mathbb{R}^{d_{\text{global}}} \tag{41}$$

$$\mathbf{B} = \mathbf{B}^{\text{SPD}} + \mathbf{B}^{\text{3D}} \qquad \in \mathbb{R}^{N \times N} \tag{42}$$

The various components of each of these equations are defined over the remainder of this section. The encoding of these features also makes recurring use of the following two generic functions. Firstly, a two-layer $\text{MLP}_{\text{encoder}}$ that projects features to a fixed-size latent space:

$$y = \text{MLP}_{\text{encoder}}(x), \quad \text{where } x \in \mathbb{R}^h \tag{43}$$

computed as

$$\bar{x} = \text{ReLU}(\text{Dense}(\text{LayerNorm}(x))) \qquad \in \mathbb{R}^{2h} \tag{44}$$

$$y = \text{Dropout}_{0.18}(\text{Dense}(\text{LayerNorm}(\bar{x}))) \quad \in \mathbb{R}^{32} \tag{45}$$

Secondly, a function $\text{Embed}_d(j) \in \mathbb{R}^d$ which selects the $j^{\text{th}}$ row from an implicit learnable weight matrix.

**Chemical Features**  The chemical features used in this work are shown in Table A.1. To embed the categorical chemical features from the dataset $\mathbb{x}_i$, $\mathbb{e}_{uv}$ into a continuous vector space, we learn a simple embedding vector for each category, sum the embeddings for all categories, and then process it with an MLP to produce $\mathbf{X}^{\text{atom}}$ and $\mathbf{E}^{\text{bond}}$, i.e.

$$\forall i: \quad \bar{\mathbf{x}}_i^{\text{atom}} = \text{MLP}_{\text{node}}\left(\sum\nolimits_{j \in \mathbb{x}_i} \text{Embed}_{64}(j)\right) \tag{46}$$

$$\forall i: \quad \mathbf{x}_i^{\text{atom}} = \text{Dropout}_{0.18}(\bar{\mathbf{x}}_i^{\text{atom}}) \in \mathbb{R}^{d_{\text{node}}} \tag{47}$$

$$\forall(u,v): \quad \bar{\mathbf{e}}_{uv}^{\text{bond}} = \text{MLP}_{\text{edge}}\left(\sum\nolimits_{j \in \mathbb{e}_{uv}} \text{Embed}_{64}(j)\right) \tag{48}$$

$$\forall(u,v): \quad \mathbf{e}_{uv}^{\text{bond}} = \text{Dropout}_{0.18}(\bar{\mathbf{e}}_{uv}^{\text{bond}}) \in \mathbb{R}^{d_{\text{edge}}} \tag{49}$$

Here $\text{MLP}_{\text{node}}$ and $\text{MLP}_{\text{edge}}$ refer to the functions by the same names used in Eq. 23 and 24 in the MPNN module, yet parameterised independently.

Table A.1: Chemical input feature selection for PCQM4Mv2.

| Node features | Feature Set | | | | |
|---|---|---|---|---|---|
| | Original | Set 1 | Set 2 | Set 3 | Ying21 |
| Atomic number | ✓ | ✓ | ✓ | ✓ | ✓ |
| Group | · | ✓ | ✓ | ✓ | · |
| Period | · | ✓ | ✓ | ✓ | · |
| Element type | · | ✓ | ✓ | ✓ | · |
| Chiral tag | ✓ | · | · | · | ✓ |
| Degree | ✓ | ✓ | ✓ | ✓ | ✓ |
| Formal charge | ✓ | ✓ | · | ✓ | ✓ |
| # Hydrogens | ✓ | ✓ | ✓ | ✓ | ✓ |
| # Radical electrons | ✓ | ✓ | ✓ | ✓ | ✓ |
| Hybridisation | ✓ | · | ✓ | ✓ | ✓ |
| Is aromatic | ✓ | ✓ | ✓ | · | ✓ |
| Is in ring | ✓ | ✓ | ✓ | ✓ | ✓ |
| Is chiral center | · | ✓ | ✓ | ✓ | ✓ |
| Explicit valence | · | · | · | · | ✓ |
| Implicit valence | · | · | · | · | ✓ |
| Total valence | · | · | · | · | ✓ |
| Total degree | · | · | · | · | ✓ |
| Default valence | · | · | · | · | ✓ |
| # Outer elecrons | · | · | · | · | ✓ |
| Van der Waals radius | · | · | · | · | ✓ |
| Covalent radius | · | · | · | · | ✓ |
| # Bonds in radius $N = 2 : 8$ | · | · | · | · | ✓ |
| Gasteiger charge | · | · | · | · | ✓ |
| Is donor | · | · | · | · | ✓ |
| Is acceptor | · | · | · | · | ✓ |
| **Edge features** | | | | | |
| Bond type | ✓ | ✓ | · | · | ✓ |
| Bond stereo | ✓ | ✓ | ✓ | ✓ | ✓ |
| Is conjugated | ✓ | · | ✓ | ✓ | ✓ |
| Is in ring | · | ✓ | ✓ | ✓ | ✓ |
| Bond direction | · | · | · | · | ✓ |

**Graph Laplacian Positional Encodings** (Kreuzer et al., 2021; Dwivedi & Bresson, 2020)   Given a graph with adjacency matrix $\mathbf{A}$ and degree matrix $\mathbf{D}$, the eigendecomposition of the graph Laplacian $\mathbf{L}$ is formulated into a global positional encoding as follows.

$$\forall i: \quad \mathbf{x}_i^{\mathrm{LapVec}} = \mathtt{MLP}_{\mathrm{encoder}}(\mathbf{U}\left[i, 2\ldots k^{\mathrm{Lap}}\right]) \in \mathbb{R}^{32},$$
$$\text{where } \mathbf{L} = \mathbf{D} - \mathbf{A} = \mathbf{U}^\top \mathbf{\Lambda} \mathbf{U} \tag{50}$$

$$\forall i: \quad \mathbf{x}_i^{\mathrm{LapVal}} = \mathtt{MLP}_{\mathrm{encoder}}\left(\frac{\mathbf{\Lambda}'}{||\mathbf{\Lambda}'||}\right) \in \mathbb{R}^{32},$$
$$\text{where } \mathbf{\Lambda}' = \mathrm{diag}(\mathbf{\Lambda})\left[2\ldots k^{\mathrm{Lap}}\right] \tag{51}$$

To produce fixed shape inputs despite variable numbers of eigenvalues / eigenvectors per graph, we truncate / pad to the lowest 7 eigenvalues, excluding the first trivial eigenvalue $\Lambda_{11} = 0$. We also randomise the eigenvector sign every epoch which is otherwise arbitrarily defined.

**Random Walk Structural Encoding** (Dwivedi et al., 2022)   This feature captures the probability that a random graph walk starting at node $i$ will finish back at node $i$, and is computed using powers of the transition matrix $\mathbf{P}$. This feature captures information about the local structures in the neighbourhood around each node, with the degree of locality controlled by the number of steps. For this submission random walks from 1 up to $k^{\mathrm{RW}} = 16$ steps were computed to form the feature vector.

$$\forall i: \quad \bar{\mathbf{x}}_i^{\mathrm{RW}} = \left[(\mathbf{P}^1)_{ii}, (\mathbf{P}^2)_{ii}, \cdots, (\mathbf{P}^{k^{\mathrm{RW}}})_{ii}\right]$$
$$\text{where } \mathbf{P} = \mathbf{D}^{-1}\mathbf{A} \tag{52}$$
$$\forall i: \quad \mathbf{x}_i^{\mathrm{RW}} = \mathtt{MLP}_{\mathrm{encoder}}\left(\bar{\mathbf{x}}_i^{\mathrm{RW}}\right) \in \mathbb{R}^{32} \tag{53}$$

**Local Graph Centrality Encoding** (Ying et al., 2021a; Shi et al., 2022)   The graph centrality encoding is intended to allow the network to gauge the importance of a node based on its connectivity, by embedding the degree (number of incident edges) of each node into a learnable feature vector.

$$\forall i: \quad \mathbf{x}_i^{\mathrm{Cent}} = \mathtt{Embed}_{64}\left(D_{ii}\right) \in \mathbb{R}^{64} \tag{54}$$

**Shortest Path Distance Attention Bias**   Graphormer (Ying et al., 2021a; Shi et al., 2022) showed that graph topology information can be incorporated into a node transformer by adding learnable biases to the self-attention matrix depending on the distance between node pairs. During data preprocessing the SPD map $\Delta \in \mathbb{N}^{N \times N}$ is computed where $\Delta_{ij}$ is the number of edges in the shortest continuous path from node $i$ to node $j$. During training each integer distance is embedded as a scalar attention bias term to create the SPD attention bias map $\mathbf{B}^{\mathrm{SPD}} \in \mathbb{R}^{N \times N}$.

$$\forall i, j: \quad B_{ij}^{\mathrm{SPD}} = \mathtt{Embed}_1\left(\Delta_{ij}\right) \in \mathbb{R} \tag{55}$$

Single-headed attention is assumed throughout this report for simplified notation, however, upon extension to multi-headed attention, one bias is learned per distance per head.

**Embedding 3D Distances**   Using the 3D positional information provided by the dataset comes with a number of inherent difficulties. Firstly, the task is invariant to molecular rotations and translations, however, the 3D positions themselves are not. Secondly, the 3D conformer positions are only provided for the training data, not the validation or test data. To deal with these two issues and take advantage of the 3D positions provided we follow the approach of Luo et al. (2022).

To ensure rotational and translational invariance we use only the distances between atoms, not the positions directly. To embed the scalar distances into vector space $\mathbb{R}^K$ we first apply $K = 128$ Gaussian kernel functions, where the $k^{\mathrm{th}}$ function is defined as

$$\forall i, j: \quad \bar{\psi}_{ij}^k = \frac{||\mathbf{r}_i - \mathbf{r}_j|| - \mu^k}{|\sigma^k|} \tag{56}$$

$$\forall i, j: \quad \psi_{ij}^k = -\frac{1}{\sqrt{2\pi}\,|\sigma^k|} \exp\left(-\frac{1}{2}\left(\bar{\psi}_{ij}^k\right)^2\right) \in \mathbb{R} \tag{57}$$

with learnable parameters $\mu^k$ and $\sigma^k$. The $K$ elements are concatenated into vector $\boldsymbol{\psi}_{ij}$. We then process these distance embeddings in three ways to produce attention biases, node features and edge features.

**3D Distance Attention Bias**   The 3D attention bias map $\mathbf{B}^{3D} \in \mathbb{R}^{N \times N}$ allows the model to modulate the information flowing between two node representations during self-attention based on the spatial distance between them, and are calculated as per Luo et al. (2022)

$$\forall i, j: \quad \mathbf{B}_{ij}^{3D} = \mathtt{MLP}_{\text{bias3D}}(\boldsymbol{\psi}_{ij}) \in \mathbb{R} \tag{58}$$

Upon extension to multi-headed attention with 32 heads $\mathtt{MLP}_{\text{bias3D}}$ instead projects to $\mathbb{R}^{32}$.

**Bond Length Encoding**   Whilst $\mathbf{B}^{3D}$ makes inter-node distance information available to the self-attention module in a dense all-to-all manner as a matrix of simple scalar biases, we also make this information available to the $\mathtt{MPNN}$ module in a sparse but high-dimensional manner as edge features $\mathbf{E}^{3D} = \left[\mathbf{e}_{uv}^{3D} \text{ for } (u,v) \in \mathcal{E}\right]$ calculated as

$$\forall (u, v): \quad \mathbf{e}_{uv}^{3D} = \mathtt{MLP}_{\text{encoder}}(\boldsymbol{\psi}_{uv}) \in \mathbb{R}^{32} \tag{59}$$

**Global 3D Centrality Encoding**   The 3D node centrality features $\mathbf{X}^{3D} = \left[\mathbf{x}_1^{3D}; \ldots; \mathbf{x}_N^{3D}\right]$ are computed by summing the embedded 3D distances from node $i$ to all other nodes. Since the sum commutes this feature cannot be used to determine the distance to a specific node, so serves as a centrality encoding rather than a positional encoding.

$$\forall i: \quad \mathbf{x}_i^{3D} = \mathbf{W}^{3D} \sum_{j \in \mathcal{V}} \boldsymbol{\psi}_{ij} \in \mathbb{R}^{32} \tag{60}$$

Here $\mathbf{W}^{3D} \in \mathbb{R}^{K \times 32}$ is a linear projection to the same latent size as the other encoded features.

**3D Denoising**   We also closely follow the approach of Luo et al. (2022) to implement an auxiliary self-supervised 3D denoising task during training. Before the 3D distance inputs are embedded (Equation 56) the atom positions $\mathbf{R}$ are substituted by $\mathbf{R} + \sigma\boldsymbol{\epsilon}$, where $\sigma \in \mathbb{R}$ is a scaling factor we set to 0.2, and $\boldsymbol{\epsilon} \in \mathbb{R}^{N \times 3}$ are Gaussian noise vectors. GPS++ computes $\hat{\boldsymbol{\epsilon}}_{ik}$, the predicted value of $\boldsymbol{\epsilon}_{ik}$, as:

$$\hat{\boldsymbol{\epsilon}}_{ik} = \left(\sum_{j \in \mathcal{V}} \mathbf{A}_{ij} \boldsymbol{\Delta}_{ij}^k \mathbf{X}_j^L \mathbf{W}_{V_1}^{3D}\right) \mathbf{W}_{V_2}^{3D} \in \mathbb{R} \tag{61}$$

where

$$\bar{\mathbf{A}} = \frac{\left(\mathbf{X}^L \mathbf{W}_Q^{3D}\right)\left(\mathbf{X}^L \mathbf{W}_K^{3D}\right)^\top}{\sqrt{d_{\text{node}}}} + \mathbf{B} \in \mathbb{R}^{N \times N} \tag{62}$$

$$\mathbf{A} = \mathtt{Dropout}_{0.3}\left(\mathtt{Softmax}\left(\bar{\mathbf{A}}\right)\right) \in \mathbb{R}^{N \times N} \tag{63}$$

Here $\mathbf{X}^L$ is the node output of the final GPS++ layer, $\boldsymbol{\Delta}_{ij}^k$ is the $k$-th element of directional vector $\frac{\mathbf{r}_i - \mathbf{r}_j}{||\mathbf{r}_i - \mathbf{r}_j||}$ from atom $j$ to atom $i$, and $\mathbf{W}_Q^{3D}, \mathbf{W}_K^{3D}, \mathbf{W}_{V_1}^{3D} \in \mathbb{R}^{d_{\text{node}} \times d_{\text{node}}}, \mathbf{W}_{V_2}^{3D} \in \mathbb{R}^{d_{\text{node}} \times 1}$ are learnable weight matrices. A cosine similarity loss is computed between $\hat{\boldsymbol{\epsilon}}$ and $\boldsymbol{\epsilon}$, and we set the loss weight ratio of Homo-Lumo MAE vs 3D Denoising to $10:1$.

# B   Hardware and Acceleration

## B.1   Hardware

We train our models using a BOW-POD16 which contains 16 IPU processors, delivering a total of 5.6 petaFLOPS of float16 compute and 14.4 GB of in-processor SRAM which is accessible at an aggregate

bandwidth of over a petabyte per second. This compute and memory is then distributed evenly over 1472 tiles per processor. This architecture has two key attributes that enable high performance on GNN and other AI workloads (Bilbrey et al., 2022): memory is kept as close to the compute as possible (i.e., using on-chip SRAM rather than off-chip DRAM) which maximises bandwidth for a nominal power budget; and compute is split up into many small independent arithmetic units meaning that any available parallelism can be extremely well utilised. In particular this enables very high performance for sparse communication ops, like gather and scatter, and achieves high FLOP utilisation even with complex configurations of smaller matrix multiplications. Both of these cases are particularly prevalent in MPNN structures like those found in GPS++.

To exploit the architectural benefits of the IPU and maximise utilisation, understanding the program structure ahead of time is key. This means all programs must be compiled end-to-end, opening up a range of opportunities for optimisation but also adding the constraint that tensor shapes must be known and fixed at compile time.

## B.2    Batching and Packing

To enable fixed tensor sizes with variable sized graphs it is common to *pad* the graphs to the max node and edge size in the dataset. This, however, can lead to lots of compute being wasted on padding operations, particularly in cases where there are large variations in the graph sizes. To combat this it is common to *pack* a number of graphs into a fixed size shape to minimise the amount of padding required, this is an abstraction that is common in graph software frameworks like PyTorch Geometric (Fey & Lenssen, 2019) and has been shown to achieve as much as 2x throughput improvement for variable length sequence models (Krell et al., 2021). Packing graphs into one single large pack, however, has a couple of significant downsides: the memory and compute complexity of all-to-all attention layers is $\mathcal{O}(n^2)$ in the pack size not the individual graph sizes, and allowing arbitrary communication between all nodes in the pack forces the compiler to choose sub-optimal parallelisation schemes for the gather/scatter operations.

To strike a balance between these two extremes we employ a two tiered hierarchical batching scheme that packs graphs into a fixed size but then batches multiple packs to form the micro-batch. We define the maximum pack size to be 60 nodes, 120 edges and 8 graphs then use a simple streaming packing method where graphs are added to the pack until either the total nodes, edges or graphs exceeds the maximum size. This achieves 87% packing efficiency of the nodes and edges with on average 3.6 graphs per pack, though we believe that this could be increased by employing a more complex packing strategy (Krell et al., 2021). We then form micro-batches of 8 packs which are pipelined (Huang et al., 2018) over 4 IPUs accumulating over 8 micro-batches and replicated 4 times to form a global batch size of 921 graphs distributed over 16 IPUs. For the MPNN-only model, the micro-batch size was increased to 15, forming global batches of 1737 graphs.

## B.3    Numerical Precision

To maximise compute throughput and maximise memory efficiency it is now common practice to use lower precision numerical formats in deep learning (Micikevicius et al., 2017). On IPUs using float16 increases the peak FLOP rate by 4x compared to float32 but also makes more effective usage of the high bandwidth on-chip SRAM. For this reason we use float16 for nearly all[1] compute but also use float16 for the majority[2] of the weights, this is made possible, without loss of accuracy, by enabling the hardware-level stochastic rounding of values. While we do use a loss scaling (Micikevicius et al., 2017) value of 1024 we find that our results are robust to a wide range of choices. We also use float16 for the first-order moment in Adam but keep the second-order moment in float32 due to the large dynamic range requirements of the sum-of-squares.

---

[1] A few operations like the sum of squares the variance calculations are up-cast to float32 by the compiler.
[2] A small number of weights are kept in float32 for simplicity of the code rather than numerical stability.

