# OpenReview forum: "GPS++: Reviving the Art of Message Passing for Molecular Property Prediction"
_TMLR — Accepted by TMLR_

### Review · Reviewer_uvAP · 2023-06-08

**Summary Of Contributions:**

This paper extends GPS. The proposed method GPS++ is a hybrid model that combines message-passing neural networks and graph transformer models. The authors tackled molecular property prediction. The proposed method achieved state-of-the-art performance. Lastly, the authors provide extensive ablation studies to investigate the effects of various features, structural encodings, and architectural components.

**Audience:**

Yes

**Broader Impact Concerns:**

I have no ethical concerns.

**Claims And Evidence:**

Yes

**Requested Changes:**

1. ~~Overall the paper reads well. However, as mentioned above, the main contribution/difference between GPS and the proposed model needs to be discussed more clearly.~~ [addressed by author feedback, Section 4. in the revised version]
2. ~~Given graph laplacian, eig vec, eig vals can be used for positional embeddings or features. However, it is not clear how to compute them for heterogeneous graphs with various edge/node types. Did you convert the heterogenous graphs into a homogeneous graph and compute $X^\text{LapVec}$ and $X^\text{LapVal}$?~~ [addressed by author feedback] treated as a homogeneous graph
3. ~~In Eq (7), why is $\sum_{(i,v) \in E}x_v^l$ included? Often, message passing along reversed edges improve the performance.~~ [addressed by author feedback]
4. ~~The feature set Ying21 is missing in Table A.1~~. [addressed by author feedback]

**Strengths And Weaknesses:**

This paper studied how to optimize the input features to improve the performance of GPS. The proposed method achieved state-of-the-art performance with an interesting hybrid architecture that combines transformers and message-passing neural networks, which is a type of graph neural networks.
The paper reads well, and the discussion in the paper is straightforward. The authors introduced the augmented general powerful scalable graph transformer in Fig 1., the new layer GPS++ layer, and feature sets.
An extensive ablation study supports the value of each component in the proposed method.
However, it is not clear which component is newly proposed in this paper. The authors did not explicitly compare the proposed method with the previous method/baseline GPS regarding neural network architectures.
In addition, in Table 2, the main experimental results raise a couple of concerns. First, in the experimental setting without 3D positional information, GPS has a twice smaller number of parameters than GPS++. This is a huge difference. Also, the proposed Hybrid method GPS++ GPS++ [No 3D Position] with more parameters underperforms GPS++ [MPNNonly, No 3D Position].
Although the proposed method achieved considerable performance gain in the setting with limited information, with 3D positional information, the performance gain is marginal, and GPS is not even compared.
In sum, the proposed extension has limited novelty, and performance gain is questionable.

**Strengths**:

1. The proposed method achieved strong performance and good efficiency by extensive optimization for architectural components, features, and structural encodings.
2. Extensive ablation study is provided and each component in the final model is well justified by empirical performance gain.

**Weaknesses**:

1. Although this paper admitted that the proposed method is a specific implementation of GPS, no clear comparison between GPS and GPS++ layers. So, it is hard to evaluate the contributions of this paper unless the differences are explicitly discussed.
2. This paper is incremental and the method section mainly introduces the implementation details including feature engineering. It may be crucial for improving performance but beyond the specific benchmarks the feature engineering has limited academic merit.
3. ~~In the setting with 3D positional information, the performance gain is marginal. Also, in Table 2, compared to GPS, GPS++ seems to improve the performance but the model size is doubled. So, it may not be a fair comparison. The authors need to provide well-tuned GPS with a comparable number of parameters~~. [addressed by authors' feedback:GPS++ (8 layers)]
4. Table 2, GPS is missing in with 3D Positional information.

---

> ### Author Response · Authors · 2023-06-23
>
> **Foreword for All Reviewers**
>
> We would like to sincerely thank all reviewers for their valuable responses to our submission. We were glad to see that clear presentation, strong empirical results and extensive ablations were consistently listed as strengths. We identified novelty and generalisability as the two main concerns, so would like to discuss these here in a common response to all reviewers before addressing more specific points from each reviewer in additional individual responses. We have submitted a new revision of the text with all the described changes, plus a version in the supplementary materials that highlights all changes in blue.
>  - **Generalisability:** All reviewers noted that results were limited to just the PCQM4Mv2 dataset, raising questions about generalisability.
> 	 - We agree that it is important to understand performance on other data and tasks. We have implemented fine-tuning of GPS++ on 12 tasks in the QM9 dataset (which differs from PCQM4Mv2 by having 3D features available at test time), and included these results in Section 6 of the revised submission under the heading Model Generalisation. GPS++ sets new state-of-the-art scores for 3 tasks in the QM9 dataset, and is otherwise broadly competitive, especially against other models that lack task-specific prediction heads such as Transformer-M which it outperforms on 7 out of 12 tasks.
> 	 - We also note that our focus on PCQM4Mv2 is a reflection of the same focus in existing graph transformer literature [Graphormer, Transformer-M, GEM-2] and public challenges [OGB: https://ogb.stanford.edu/], where its size (3.7M molecules vs QM9’s 134K) makes it one of the largest publicly available datasets of its kind and uniquely valuable for training models with high data requirements.
>  - **Novelty:** Many reviewers suggested this submission lacks sufficient novelty, though reviewer T4FP also noted that the TMLR guidelines explicitly advise against judgements on novelty, focussing instead on whether there is sufficient interest from other researchers in the field. To this end we believe this submission has the following valuable contributions that will be of interest:
> 	 - Achieving SOTA results on a highly competitive standard benchmark dataset using a parametrically efficient model with open-sourced code and checkpoints.
> 	 - Extensive ablations measuring the empirical value of a broad range of feature engineering, architecture design, loss function and regularisation choices from across the recent molecular property prediction literature.
> 	 - Challenging the current trend towards graph transformers being the only performant architecture for molecular property prediction, and encouraging renewed interest in Message Passing networks which have recently fallen out of favour. We believe our MPNN-only model achieving parity with the prior SOTA Transformer-M will be unexpected to some researchers who may have assumed MPNNs are no longer competitive in this setting.
>  - **Changes from GPS:**  Some reviewers requested a clearer explanation of the changes between GPS and GPS++. We have now added a direct comparison at the end of Section 4, which highlights the following points:
> 	 - Replacing the GatedGCN with a custom and highly-tuned MPNN module, most similar to Battaglia et. al. (2018) but with some key differences explained in Section 4.1 (namely adjacent node aggregation, aggregating messages back into the sending node, and using varied hidden sizes for node/edge/global features)
> 	 - Incorporating the 2D spatial attention bias (SPD bias) and centrality encoding from Graphormer (Ying et. al. 2021)
> 	 - Following the approach of Transformer-M to incorporate training-only 3D inputs as node features, edge features and attention biases (via grouped input masking), as well as the SE3-equivariant 3D denoising head + loss term.
> 	 - The noisy nodes/edges regularisation loss from Godwin et. al. (2022)
> 	 - Original chemical atom features (called Group, Period & Type) that we believe may be broadly useful for any work attempting to generalise to atoms with unseen atomic numbers.
> 	 - Stochastic depth (Huang et.al 2016)
> 	 - Rigorous hyper-parameter tuning
>
> We look forward to hearing your thoughts on these comments and changes.

---

> ### Author Response · Authors · 2023-06-23
>
> **Response to Review 4 uvAP**
>
> Thank you for your review, we’re glad you think it reads well and that our choices of components were well justified.
>
> We agree with your suggestion that we have not provided a fair comparison between GPS++ and the prior GPS result. Our ablation section shows how much each of our components contribute to task performance, but the exact differences were not made clear in the text so we have added a comparison to the end of Section 4. After your feedback we have also experimented with halving the number of GPS++ layers to create a more parametrically comparable `GPS++ (8 layers)` result as follows:
>
> $
> \\begin{array} {lcr}
> \\hline
> \textbf{Model} & \textbf{Parameters} & \textbf{Valid MAE (meV)} \\\\
> \\hline
> \text{Without 3D Positions} &  &  \\\\
> \\hline
> \text{GPS}  & 19.4M   & 85.8 \\\\
> \text{GPS++ (8 layers)} & 22.4M   & \textbf{(New) } 83.6 \\\\
> \text{GPS++ (8 layers, MPNN only)} & 20.3M   & \textbf{(New) } 82.2 \\\\
> \text{GPS++ (MPNN only)} & 40.0M  & 81.8 \\\\
> \\hline
> \text{With 3D Positions} &  &  \\\\
> \\hline
> \text{GPS++ (8 layers, MPNN only)}  & 22.5M  & \textbf{(New) } 79.3 \\\\
> \text{GPS++ (8 layers)}  & 22.7M   & \textbf{(New) } 78.6 \\\\
> \text{Transformer-M}  & 69.0M  & 77.2 \\\\
> \text{GPS++} & 44.5M   & \textbf{76.6} \\\\
> \\hline
> \\end{array}
> $
>
> These have been added to the main results table (Table 2) in the revised submission.
>
> To stay aligned with the core message of the paper, however, we would like to keep Transformer-M rather than GPS as the main point of comparison. Recent progress on molecular property prediction, particularly PCQM4Mv2 (which has become a defacto benchmark due to its size), has been dominated by graph transformer models, to the extent that a researcher looking at the leaderboard may conclude transformers have an intrinsic architectural advantage on this task over message passing. Our submission challenges this narrative by rigorously reimplementing all the feature-engineering, multi-task loss design and architectural / hyper-parameter optimisation work that has implicitly accumulated in the work on graph transformers, and shows that when an MPNN-based model is given the same engineering effort it can be highly competitive (and even set the SOTA). This is why we put a strong emphasis on our MPNN-only results. We believe direct comparison with Transformer-M is particularly meaningful and informative since we have integrated many of their innovations in 3D positional encodings, 3D denoising and regularisation, and therefore have exact counterparts to their ablation results. This cross-architecture comparison may be of more interest to the academic community than our improvement over the reference implementation of the GPS framework.
>
> Regarding your concern that our MPNN-only model outperforms the full hybrid GPS++ in the 2D setting, we don’t feel this detracts from the success of the hybrid in the 3D setting. Rather, we believe the contrast in behaviours between these two settings is very informative, in particular for those researchers interested in the differences between MPNNs and graph transformers. Specifically, it implies that the Transformer component is highly reliant on high-quality 3D positional information to learn spatial relations (despite having a 2D attention bias term), whilst the MPNN can make good use of the spatial priors encoded into the model by passing messages along chemical bonds. Perhaps the more weakly conditioned Transformer could learn these relationships with enough data, raising a possible need for much larger datasets of this kind.
>
> **Requested Changes:**
>  - As described in the shared response to all reviewers, we have updated the text to explicitly compare GPS++ to GPS.
>  - That is correct, we consider PCQM4Mv2 to be a homogeneous graph task, where the graph laplacian encodes only positional/structural information and the node/edge types are provided as input features to the encoder layers.
>  - In our case, $\sum_{(i, v)\in E} \mathbf{x}^l_v$ is not included because we represent every molecular bond with a pair of oppositely oriented edges, meaning that the sum over the 1-hop neighbourhood of incoming edges is the same as the sum over the 1-hop neighbourhood of outgoing edges.
>  - We have updated Table A.1 to include the Ying21 feature set, thank you for highlighting this.
>
> We hope we have addressed your key concerns with this submission, please let us know your thoughts.

---

> > ### Comment · Reviewer_uvAP · 2023-06-26
> > **I read author feedback**
> >
> > Thank you for the detailed author feedback with additional experimental results. This addresses one of my concerns. Although the authors added explicit comparisons with GPS, the key differences/technical contributions are not significant except for implementation details.

---

> ### Author Response · Authors · 2023-06-29
>
> Thank you for your feedback, we are sorry to hear you don’t find our contributions significant.
>
> Though you haven’t crossed weakness 1 from your original review, we believe we have addressed it by adding the explicit GPS comparison section.
>
> To comment further on weakness 4, the original GPS model does not produce results using 3D input features, relying instead on 2D graph positional encodings across every dataset they test. We have attempted to include the next best thing for the sake of the comparison, namely we have included a version of GPS++ that has no 3D features in the main results table, allowing direct comparison with the no-3D GPS result.

---

### Review · Reviewer_9ysq · 2023-06-10

**Summary Of Contributions:**

The authors present a hybrid graph neural network model called the GPS++, utilizing elements of both MPNNs and transformer architectures, focused on the HOMO-LUMO gap prediction task on the  PCQM4Mv2 dataset. The paper is a mainly empirical contribution, including and evaluating various features and network blocks within the GPS framework, resulting in a very good performance on the selected regression task, outperforming other state-of-the-art GNN models. The paper also includes extensive ablation studies, clearly showcasing the contribution of the individual features/network components to the overall performance, leading to the interesting finding that most of the performance can be attributed to the MPNN component as opposed to the attention component of the architecture, with the MPNN alone showing better performance on the 2d structure task.

**Audience:**

Yes

**Broader Impact Concerns:**

I have no concerns about the ethical implications of this work.

**Claims And Evidence:**

Yes

**Requested Changes:**

Overall, I think the paper is good as it is, the authors set a specific objective and achieve it, giving the paper a narrow but clear structure. While it would certainly be beneficial to include results for other similar datasets such as QM9, PDBBind, etc, which would significantly strengthen the work, I think the work as it is just about satisfies the acceptance criteria of TMLR and I would recommend it for acceptance without requiring any major changes in the content.
However, for the purposes of reproducibility, I believe it is also important for the authors to provide access to the code used to produce the results, and unless I missed it i did not see the code included as a part of the submission. Not having the code available is potentially a reason to not accept the paper, since reproducibility is even more important given the empirical nature of this work and lack of theoretical novelty.

In term of minor changes, some more details on the training procedure would be appreciated, specifically about how the auxiliary task of 3d positions denoising is incorporated into the training process. The authors give ratios of the losses for the noisy nodes/edges task during training, but no such details are given for the 3d denoising task for training.

**Strengths And Weaknesses:**

Strengths:
The main strength of this paper is the detailed empirical evaluation and the extensive ablation studies of the different design choices of the network architecture. The authors utilize the generality of the GPS framework to combine various established feature engineering methods and network components, and in doing so achieve a state-of-the-art performance on the selected task. Furthermore, the detailed ablation studies can potentially be used to guide the design decisions of researchers or engineers when encountered with a similar task in the future.
Another strength of the work is its clearness and readability, with the authors clearly stating the reason and justification for each design decision, and having a well defined notation, making the equations easy to follow.
Weaknesses:
The two main weaknesses of this work are its narrow focus and the lack of novelty.
Most of the other state-of-the-art GNN models included in the paper as a point of comparison have been evaluated on a number of graph-based tasks and datasets, showing their generality and potential for wide range application. On the other hand the GPS++ was only evaluated on the HOMO-LUMO gap prediction task of the PCQM4Mv2 dataset. Due to the large number and specificity of the components used in GPS++,  it is unclear whether its good performance generalizes to other graph-based tasks or whether its architecture is fine-tuned to provide exceptional results only for the selected task.
In addition to the narrow focus, the work also fails to present any novel theoretical contributions to the field, although to the credit of the authors they also do not claim any such contribution. The model is a specific instantiation of the GPS framework, which is clearly signaled by its name GPS++, and largely uses already established features and network components that are introduced in a lot of the competing models. As mentioned previously, the contribution mainly comes in the way that these components are combined together and the extensive ablation studies.

---

> ### Author Response · Authors · 2023-06-23
>
> **Foreword for All Reviewers**
>
> We would like to sincerely thank all reviewers for their valuable responses to our submission. We were glad to see that clear presentation, strong empirical results and extensive ablations were consistently listed as strengths. We identified novelty and generalisability as the two main concerns, so would like to discuss these here in a common response to all reviewers before addressing more specific points from each reviewer in additional individual responses. We have submitted a new revision of the text with all the described changes, plus a version in the supplementary materials that highlights all changes in blue.
>  - **Generalisability:** All reviewers noted that results were limited to just the PCQM4Mv2 dataset, raising questions about generalisability.
> 	 - We agree that it is important to understand performance on other data and tasks. We have implemented fine-tuning of GPS++ on 12 tasks in the QM9 dataset (which differs from PCQM4Mv2 by having 3D features available at test time), and included these results in Section 6 of the revised submission under the heading Model Generalisation. GPS++ sets new state-of-the-art scores for 3 tasks in the QM9 dataset, and is otherwise broadly competitive, especially against other models that lack task-specific prediction heads such as Transformer-M which it outperforms on 7 out of 12 tasks.
> 	 - We also note that our focus on PCQM4Mv2 is a reflection of the same focus in existing graph transformer literature [Graphormer, Transformer-M, GEM-2] and public challenges [OGB: https://ogb.stanford.edu/], where its size (3.7M molecules vs QM9’s 134K) makes it one of the largest publicly available datasets of its kind and uniquely valuable for training models with high data requirements.
>  - **Novelty:** Many reviewers suggested this submission lacks sufficient novelty, though reviewer T4FP also noted that the TMLR guidelines explicitly advise against judgements on novelty, focussing instead on whether there is sufficient interest from other researchers in the field. To this end we believe this submission has the following valuable contributions that will be of interest:
> 	 - Achieving SOTA results on a highly competitive standard benchmark dataset using a parametrically efficient model with open-sourced code and checkpoints.
> 	 - Extensive ablations measuring the empirical value of a broad range of feature engineering, architecture design, loss function and regularisation choices from across the recent molecular property prediction literature.
> 	 - Challenging the current trend towards graph transformers being the only performant architecture for molecular property prediction, and encouraging renewed interest in Message Passing networks which have recently fallen out of favour. We believe our MPNN-only model achieving parity with the prior SOTA Transformer-M will be unexpected to some researchers who may have assumed MPNNs are no longer competitive in this setting.
>  - **Changes from GPS:**  Some reviewers requested a clearer explanation of the changes between GPS and GPS++. We have now added a direct comparison at the end of Section 4, which highlights the following points:
> 	 - Replacing the GatedGCN with a custom and highly-tuned MPNN module, most similar to Battaglia et. al. (2018) but with some key differences explained in Section 4.1 (namely adjacent node aggregation, aggregating messages back into the sending node, and using varied hidden sizes for node/edge/global features)
> 	 - Incorporating the 2D spatial attention bias (SPD bias) and centrality encoding from Graphormer (Ying et. al. 2021)
> 	 - Following the approach of Transformer-M to incorporate training-only 3D inputs as node features, edge features and attention biases (via grouped input masking), as well as the SE3-equivariant 3D denoising head + loss term.
> 	 - The noisy nodes/edges regularisation loss from Godwin et. al. (2022)
> 	 - Original chemical atom features (called Group, Period & Type) that we believe may be broadly useful for any work attempting to generalise to atoms with unseen atomic numbers.
> 	 - Stochastic depth (Huang et.al 2016)
> 	 - Rigorous hyper-parameter tuning
>
> We look forward to hearing your thoughts on these comments and changes.

---

> ### Author Response · Authors · 2023-06-23
>
> **Response to Reviewer 9ysq**
>
> Thank you for your thorough review, we were pleased to hear that you would recommend the submission for acceptance. We are also glad that you have highlighted our MPNN-only results as interesting, since we feel this is one of our more unexpected contributions. With regard to the weaknesses identified, we have discussed both novelty and generalisability in the shared response to all reviewers.
>
> **Requested changes:**
>  - As mentioned, the revised submission includes QM9 fine-tuning results.
>  - We have open-sourced our code base; the link will appear at the end of Section 1 under “Reproducibility”, but it is currently redacted for the purpose of the double-blind review.
>  - Thank you for highlighting this omission. We have updated the text of Section A.2 that describes the 3D noise prediction head, adding the unspecified Gaussian noise scale and the loss ratio.
>
> We hope this addresses your concerns.

---

> > ### Comment · Reviewer_9ysq · 2023-06-26
> >
> > Thank you addressing many of the concerns of me and the other reviewers, I am happy with the changes and recommend this paper for publication.

---

### Review · Reviewer_T4FP · 2023-06-10

**Summary Of Contributions:**

This work proposes a deep model for graph learning, particularly predicting the HOMO-LUMO gap of molecules. It closely follows the design of a previous work GPS, which combines the operation of message passing and transformer. The main difference between this GPS++ work, compared to GPS, lies in a well-tuned MPNN model, feature engineering, and dedicatedly designed training objective.

**Audience:**

Yes

**Claims And Evidence:**

Yes

**Requested Changes:**

(1) It would be better to evaluate the GPS++ method on more tasks.

**Strengths And Weaknesses:**

#### Strengths ####

(1) This work provides exhaustive experiments to demonstrate a lot number of different model architecture designs and feature engineering. The extensive experimental results could be useful for the community for future research.

(2) The writing and the organization of the paper is super clear and easy to follow.

(3) The good empirical performance of the dedicatedly designed model, including feature engineering.

#### Weaknesses ####

(1) One of the biggest weaknesses is the novelty of the method. Since GPS++ closely follows the general design of previous models, the architecture itself is not new. Since TMLR does not require a lot of novelty, I will not criticize this aspect in terms of making an acceptance decision.

(2) The experiments are only on a single dataset and task, although the dataset is super large. Readers may wonder if the exhaustive model designs and featurization work for other molecular property prediction tasks. So I highly recommend adding at least one more experiment to make the paper stronger.

---

> ### Author Response · Authors · 2023-06-23
>
> **Foreword for All Reviewers**
>
> We would like to sincerely thank all reviewers for their valuable responses to our submission. We were glad to see that clear presentation, strong empirical results and extensive ablations were consistently listed as strengths. We identified novelty and generalisability as the two main concerns, so would like to discuss these here in a common response to all reviewers before addressing more specific points from each reviewer in additional individual responses. We have submitted a new revision of the text with all the described changes, plus a version in the supplementary materials that highlights all changes in blue.
>  - **Generalisability:** All reviewers noted that results were limited to just the PCQM4Mv2 dataset, raising questions about generalisability.
> 	 - We agree that it is important to understand performance on other data and tasks. We have implemented fine-tuning of GPS++ on 12 tasks in the QM9 dataset (which differs from PCQM4Mv2 by having 3D features available at test time), and included these results in Section 6 of the revised submission under the heading Model Generalisation. GPS++ sets new state-of-the-art scores for 3 tasks in the QM9 dataset, and is otherwise broadly competitive, especially against other models that lack task-specific prediction heads such as Transformer-M which it outperforms on 7 out of 12 tasks.
> 	 - We also note that our focus on PCQM4Mv2 is a reflection of the same focus in existing graph transformer literature [Graphormer, Transformer-M, GEM-2] and public challenges [OGB: https://ogb.stanford.edu/], where its size (3.7M molecules vs QM9’s 134K) makes it one of the largest publicly available datasets of its kind and uniquely valuable for training models with high data requirements.
>  - **Novelty:** Many reviewers suggested this submission lacks sufficient novelty, though reviewer T4FP also noted that the TMLR guidelines explicitly advise against judgements on novelty, focussing instead on whether there is sufficient interest from other researchers in the field. To this end we believe this submission has the following valuable contributions that will be of interest:
> 	 - Achieving SOTA results on a highly competitive standard benchmark dataset using a parametrically efficient model with open-sourced code and checkpoints.
> 	 - Extensive ablations measuring the empirical value of a broad range of feature engineering, architecture design, loss function and regularisation choices from across the recent molecular property prediction literature.
> 	 - Challenging the current trend towards graph transformers being the only performant architecture for molecular property prediction, and encouraging renewed interest in Message Passing networks which have recently fallen out of favour. We believe our MPNN-only model achieving parity with the prior SOTA Transformer-M will be unexpected to some researchers who may have assumed MPNNs are no longer competitive in this setting.
>  - **Changes from GPS:**  Some reviewers requested a clearer explanation of the changes between GPS and GPS++. We have now added a direct comparison at the end of Section 4, which highlights the following points:
> 	 - Replacing the GatedGCN with a custom and highly-tuned MPNN module, most similar to Battaglia et. al. (2018) but with some key differences explained in Section 4.1 (namely adjacent node aggregation, aggregating messages back into the sending node, and using varied hidden sizes for node/edge/global features)
> 	 - Incorporating the 2D spatial attention bias (SPD bias) and centrality encoding from Graphormer (Ying et. al. 2021)
> 	 - Following the approach of Transformer-M to incorporate training-only 3D inputs as node features, edge features and attention biases (via grouped input masking), as well as the SE3-equivariant 3D denoising head + loss term.
> 	 - The noisy nodes/edges regularisation loss from Godwin et. al. (2022)
> 	 - Original chemical atom features (called Group, Period & Type) that we believe may be broadly useful for any work attempting to generalise to atoms with unseen atomic numbers.
> 	 - Stochastic depth (Huang et.al 2016)
> 	 - Rigorous hyper-parameter tuning
>
> We look forward to hearing your thoughts on these comments and changes.

---

> ### Author Response · Authors · 2023-06-23
>
> **Response to Reviewer T4FP**
>
> Thank you very much for your review, your feedback is very clear and helpful. We believe we have commented on your concerns in our response to all reviewers, namely by providing new results showing generalisation to the QM9 tasks and clarifying what we believe is the core of our contribution.
> Please let us know any feedback you have on these points.

---

> > ### Comment · Reviewer_T4FP · 2023-06-27
> > **Thanks**
> >
> > Thanks for the response. My original concern has been addressed. I highly recommend doing the same experiments for GPS on QM9 and including it in the paper to further demonstrate the improvement of GPS++ over GPS.

---

> ### Author Response · Authors · 2023-06-29
>
> Thanks for the feedback, we’re pleased you feel your original concerns have been addressed. We agree it would be interesting to include a comparison to the original GPS on the same QM9 tasks. Unfortunately, these results are not available in the original GPS paper, and since we have not worked with the GPS code base before we do not have sufficient confidence in modifying it ourselves to generate fair and reliable results on a new dataset in this short discussion window.
>
> We have considered the alternative of providing QM9 results for GPS++ with ablated 3D features, attention bias terms, input features and auxiliary losses to make an approximation of the original GPS model within our codebase. However, we feel that presenting these results as representative of / directly comparable to GPS’s performance could be misleading due to many remaining differences in the MPNN architecture and general implementation. Moreover, QM9 is a fully 3D dataset (unlike PCQM4Mv2 which has no 3D features at test time) so these 2D-only results may not be meaningfully comparable to other results in the literature.
>
> Please let us know your thoughts on this.

---

### Review · Reviewer_Souw · 2023-06-11

**Summary Of Contributions:**

In this paper, the authors propose a novel graph-learning architecture designed specifically for molecular property prediction tasks. The proposed model, namely GPS++, consists of a combination of architectures/modules that exists in prior works.
The paper primarily focuses on the molecules HOMO-LUMO energy gap prediction and shows the proposed method achieves state-of-the-art results on the OGB-LSC dataset PCQM4Mv2. In addition to the main results, the paper also includes extensive ablation studies to showcase the contribution of each neural module and input features to the final results.

**Audience:**

Yes

**Broader Impact Concerns:**

I don't have ethical concerns about this work.

**Claims And Evidence:**

Yes

**Requested Changes:**

* Include empirical studies on more datasets/tasks to examine the general applicability of the GPS++ module.
* Add more discussions on the motivations and reasoning behind the design of the new neural architecture.

**Strengths And Weaknesses:**

Strengths: Although only study a very specific application in graph learning, the paper clearly defines its scope and provides enough context for the problem setup and discussion of prior works. The proposed architecture combines the graph transformers and the message-passing networks and achieves state-of-the-art results in the OGB-LSC task. The paper provides extensive ablation empirical studies on the properties of the node/edge/3D features as well as the effectiveness of the massage-passing neural module. The architecture of the proposed method is well presented and the structure of the paper is easy to follow.

Weaknesses: I have two main concerns for this work: novelty and generalizability. The proposed architecture largely resembles that of GPS, with limited modifications. Also, it is not clear the motivations behind the design of the new GPS++ block. The paper only uses a single dataset in all experiments. It is not clear whether the proposed method only achieves better performance in this specific case or it's general enough to be used in other molecular graph learning tasks.

---

> ### Author Response · Authors · 2023-06-23
>
> **Foreword for All Reviewers**
>
> We would like to sincerely thank all reviewers for their valuable responses to our submission. We were glad to see that clear presentation, strong empirical results and extensive ablations were consistently listed as strengths. We identified novelty and generalisability as the two main concerns, so would like to discuss these here in a common response to all reviewers before addressing more specific points from each reviewer in additional individual responses. We have submitted a new revision of the text with all the described changes, plus a version in the supplementary materials that highlights all changes in blue.
>  - **Generalisability:** All reviewers noted that results were limited to just the PCQM4Mv2 dataset, raising questions about generalisability.
> 	 - We agree that it is important to understand performance on other data and tasks. We have implemented fine-tuning of GPS++ on 12 tasks in the QM9 dataset (which differs from PCQM4Mv2 by having 3D features available at test time), and included these results in Section 6 of the revised submission under the heading Model Generalisation. GPS++ sets new state-of-the-art scores for 3 tasks in the QM9 dataset, and is otherwise broadly competitive, especially against other models that lack task-specific prediction heads such as Transformer-M which it outperforms on 7 out of 12 tasks.
> 	 - We also note that our focus on PCQM4Mv2 is a reflection of the same focus in existing graph transformer literature [Graphormer, Transformer-M, GEM-2] and public challenges [OGB: https://ogb.stanford.edu/], where its size (3.7M molecules vs QM9’s 134K) makes it one of the largest publicly available datasets of its kind and uniquely valuable for training models with high data requirements.
>  - **Novelty:** Many reviewers suggested this submission lacks sufficient novelty, though reviewer T4FP also noted that the TMLR guidelines explicitly advise against judgements on novelty, focussing instead on whether there is sufficient interest from other researchers in the field. To this end we believe this submission has the following valuable contributions that will be of interest:
> 	 - Achieving SOTA results on a highly competitive standard benchmark dataset using a parametrically efficient model with open-sourced code and checkpoints.
> 	 - Extensive ablations measuring the empirical value of a broad range of feature engineering, architecture design, loss function and regularisation choices from across the recent molecular property prediction literature.
> 	 - Challenging the current trend towards graph transformers being the only performant architecture for molecular property prediction, and encouraging renewed interest in Message Passing networks which have recently fallen out of favour. We believe our MPNN-only model achieving parity with the prior SOTA Transformer-M will be unexpected to some researchers who may have assumed MPNNs are no longer competitive in this setting.
>  - **Changes from GPS:**  Some reviewers requested a clearer explanation of the changes between GPS and GPS++. We have now added a direct comparison at the end of Section 4, which highlights the following points:
> 	 - Replacing the GatedGCN with a custom and highly-tuned MPNN module, most similar to Battaglia et. al. (2018) but with some key differences explained in Section 4.1 (namely adjacent node aggregation, aggregating messages back into the sending node, and using varied hidden sizes for node/edge/global features)
> 	 - Incorporating the 2D spatial attention bias (SPD bias) and centrality encoding from Graphormer (Ying et. al. 2021)
> 	 - Following the approach of Transformer-M to incorporate training-only 3D inputs as node features, edge features and attention biases (via grouped input masking), as well as the SE3-equivariant 3D denoising head + loss term.
> 	 - The noisy nodes/edges regularisation loss from Godwin et. al. (2022)
> 	 - Original chemical atom features (called Group, Period & Type) that we believe may be broadly useful for any work attempting to generalise to atoms with unseen atomic numbers.
> 	 - Stochastic depth (Huang et.al 2016)
> 	 - Rigorous hyper-parameter tuning
>
> We look forward to hearing your thoughts on these comments and changes.

---

> ### Author Response · Authors · 2023-06-23
>
> Thank you for your review, we are happy to hear you found the presentation clear and that you appreciated the extensive empirical ablations. In the foreword, we discussed your main concerns regarding novelty and generalisability. We are happy to answer further questions.
>
> **Your Requested Changes**
>  - As mentioned in the response to all reviewers, we have included results from fine-tuning on 12 tasks in the QM9 dataset, showing the GPS++ module has more general applicability than just PCQM4Mv2.
>  - We have modified the submission to better explain our architectural decisions. In particular we have extended our discussion of the non-standard configuration of our MPNN module in Section 4.1. Please let us know any other specific areas you would like to be covered in more detail.
>
> We hope you feel these answers address the issues you have raised.

---

> > ### Comment · Reviewer_Souw · 2023-06-27
> > **Thanks**
> >
> > Thanks for adding additional empirical results. I noticed that GPS++ is among some of the worst performing models for some metrics and would recommend to add more discussion on this performance gap.

---

> ### Author Response · Authors · 2023-07-14
>
> Thank you for your feedback, we have now made several changes to the text that we have described in a general comment for all reviewers [here](https://openreview.net/forum?id=moVEUgJaHO&noteId=cq6FIhknS9).
>
> We hope you feel this addresses your request.

---

### Author Response · Authors · 2023-07-14
**Comment for all reviewers**

Thank you for your patience during our extended discussion period, we have used this time to further build our confidence in the new QM9 results, as well as improve several of the scores and expand on the discussion around GPS++’s strengths and weaknesses as requested by reviewer Souw.

Firstly, our original revision featured a very slim table of results from the existing literature for comparison, intended just to capture the state of the art for each label. We have now expanded this table to include a much broader selection of results; these convey a clearer sense of the distribution of scores across many methods from the literature, in particular for those labels where GPS++ appeared weakest (R2 and ZPVE). Moreover, we have included new results from GNS-TAT+NN Zaidi et al. (2023) which improve the SOTA on the Homo, Lumo and Gap labels versus the previous best from Transformer-M.

Secondly, as part of the process of investigating GPS++’s relative strengths and weaknesses on different tasks we have refined our choices of more fine-tuning hyperparameters (beyond the original learning rate and epoch sweep) to account for per-label differences in behaviour. In particular, we found the roles of our 3D denoising auxiliary loss function and label normalisation are quite variable across tasks, and we have detailed this in our updated discussion. Our improved results are shown in the updated Table 3: we highlight that both R2 and ZPVE have improved over the results we provided previously (R2 by more than 2.5x).

Finally, we have put considerable effort into investigating the energy labels U0, U, H and G (which are very closely related to one another). We have found that unfortunately the scores that we listed for these labels in our previous text revision used incorrect units, comparing our MAEs in Electron Volts (eV) to prior results which are in Millielectron Volts (meV). Therefore, our results were out by 3 orders of magnitude and only coincidentally appeared reasonable compared to prior literature. After correcting the units, we found improving these scores difficult until attempting a per-atom energy normalisation method described in [here](https://schnetpack.readthedocs.io/en/stable/tutorials/tutorial_02_qm9.html) that afforded an immediate 500x improvement. This normalisation appears to be key to performance on these tasks, but is often not mentioned in the literature or present in the published code, making reproduction of and comparison to prior results difficult. Following this step change in test accuracy, for the remainder of our extension period we have continued to improve performance on the label U by tuning hyperparameters to the new training dynamics, and are already reaching scores much more comparable with prior work. Despite current progress, our results exhibit very high variance and sensitivity to hyperparameters, so we believe there is a lot more work that could be done on U and its accompanying labels. For this reason, and due to the limited information around prior methodologies, we are still not confident these represent meaningful comparison points without further exploration. Rather than continuing to delay the review process working on U and subsequently H, G and U0, we have opted to follow the Action Editor’s advice and reduce the scope of the QM9 fine-tuning task, omitting these labels from the study. We believe the remaining 8 tasks still present a representative view of the model’s generalisation performance, highlighting GPS++’s relative strengths (Homo, Lumo, Gap, and Mu) and weaknesses (ZPVE and R2).

Thank you again for your patience, we look forward to hearing your thoughts.

---

### Decision · Action_Editors · 2023-06-28

**Recommendation:** Accept as is

**Comment:**

Both reviewers and authors engaged in a fruitful discussion that resulted in substantial improvements of the paper.

All reviewers like the paper especially after the revisions but are also concerned about the lack of changes over the original GPS method. In the end the reviewers were convinced that the changes merit publication.

**Audience:**

Yes

**Claims And Evidence:**

Yes